# Bromo- and iodo-bridged building units in metal-organic frameworks for enhanced carrier transport and CO$_2$ photoreduction by water vapor

Xinfeng Chen[1], Chengdong Peng[1], Wenyan Dan[1], Long Yu[1], Yinan Wu[2,3] & Honghan Fei [1] ✉

Organolead halide hybrids have many promising attributes for photocatalysis, *e.g.* tunable bandgaps and excellent carrier transport, but their instability constraints render them vulnerable to polar molecules and limit their photocatalysis in moisture. Herein, we report the construction of metal−organic frameworks based on [Pb$_2$X]$^{3+}$ (X = Br−/I−) chains as secondary building units and 2-amino-terephthalate as organic linkers, and extend their applications in photocatalytic CO$_2$ reduction with water vapor as the reductant. Hall effect measurement and ultrafast transient absorption spectroscopy demonstrate the bromo/iodo-bridged frameworks have substantially enhanced photocarrier transport, which results in photocatalytic performances superior to conventional metal-oxo metal-organic frameworks. Moreover, in contrast to lead perovskites, the [Pb$_2$X]$^{3+}$-based frameworks have accessible porosity and high moisture stability for gas-phase photocatalytic reaction between CO$_2$ and H$_2$O. This work significantly advances the excellent carrier transport of lead perovskites into the field of metal-organic frameworks.

Developing highly efficient, selective and stable catalysts for reduction of CO$_2$ into high-value chemicals under the mild conditions is essential to realize a carbon-neutral cycle and address the global energy crisis[1–3]. Among the vast majority of photo(electro)catalytic systems, the artificial photosynthesis to use inexhaustible solar energy to achieve the simultaneous CO$_2$ photoreduction and H$_2$O oxidation into value-added products has attracted increasing attentions[4–7]. However, using H$_2$O as the sacrificial agent for CO$_2$ photoreduction will inevitably lead to the competition with the side reaction of proton reduction[8,9]. To address this scenario, a number of synthetic strategies such as junction-component incorporation (i.e. fabrication of Z-Scheme or S-Scheme photocatalysts) and facet/defect engineering have been studied to improve the existing photocatalytic systems[10–12].

Nevertheless, it still remains a great challenge to develop an intrinsic, single-phase photocatalyst to realize the efficient, selective CO$_2$ reduction by water vapor.

Metal-organic frameworks (MOFs), an emerging class of crystalline porous solids, have been developed to be one of the most efficient and promising CO$_2$ adsorbents, therefore affording substantially enhanced local concentration of CO$_2$ in porosity[13]. In terms of photocatalytic CO$_2$ reduction, the competitive proton reduction reaction has a much higher potential barrier in gas phase (5.39 eV, H$_2$O to H$_2$) than in liquid phase (1.25 eV, H$^+$ to H$_2$)[14]. The well-defined porosity endows MOFs to be an excellent platform for the fast mass transport of gas-phase molecules and potentially efficient photocatalysis. Deng and co-workers have discovered the internal growth of TiO$_2$ inside a

[1]School of Chemical Science and Engineering, Shanghai Key Laboratory of Chemical Assessment and Sustainability, Tongji University, Shanghai 200092, PR China. [2]College of Environmental Science and Engineering, State Key Laboratory of Pollution Control and Resource Reuse, Tongji University, Shanghai 200092, PR China. [3]Shanghai Institute of Pollution Control and Ecological Security, Shanghai 200092, PR China. ✉e-mail: fei@tongji.edu.cn

mesoporous MOF to generate molecular compartments, achieving efficient gas-phase $CO_2$ photoreduction in the presence of water vapor[15]. Recently, Cao and co-workers have also synthesized a porphyrin-based Ni-MOF with high gas uptake capacity and unique coordination environment for gas-phase $CO_2$ photoreduction with $H_2O$[16].

The vast majority of MOFs are constructed by photocatalytically active metal-oxo clusters (namely secondary building units, SBUs), which are often limited by narrow light absorption. For example, the UV-absorbing $Zr_6O_4(OH)_4$ clusters are the active sites for $CO_2$ photoreduction in the benchmark UiO-series MOFs[17]. The introduction of light-harvesting organic linkers and/or guest molecules are necessary to achieve visible-light-driven photocatalysis[18,19]. However, the conventional metal-oxo SBUs in carboxylate-based MOFs have very confined carrier mobility and charge transport properties[20,21]. Very few MOFs have been reported to occupy halide-bridged SBUs, until the very recent discovery of UiO-66 type MOFs with fluoro bridging groups[22].

In contrast to the metal oxide counterpart, metal halide hybrids are an emerging class of photoactive materials with tunable bandgap, high light absorption coefficient, and excellent charge transport property, showing great potentials in photovoltaics and photocatalysis[23]. However, the insufficient chemical stability of many metal halide hybrids has limited their photocatalytic applications by using $H_2O$ as the sacrificial reductant. The vast majority of their uses in $CO_2$ photoreduction were performed in organic solvents (*e.g.* ethyl acetate, acetonitrile)[24,25]. To overcome this challenge, researchers have stabilized the metal halide hybrids in different solid-state matrix (*e.g.* graphene oxide, g-$C_3N_4$, MOFs), despite the complicated synthesis and limited long-term stability[26–28]. Very recently, our group have reported an organolead iodide crystalline material for photocatalytic overall water splitting, but the densely packed layered structure has no accessible porosity for gas-phase reaction[29].

Herein, we realize the bridging between two distinctive classes of solid-state materials, i.e. MOFs and organolead halide hybrids, by crystal engineering. The resultant open frameworks consist of 1D $[Pb_2Br]^{3+}$ or $[Pb_2I]^{3+}$ chains as SBUs and 2-aminoterephthalic acid ($NH_2$-bdc) as linkers, combining the advantageous properties of both MOFs

and lead halide hybrids. Unlike the ionic structures of conventional organolead halide perovskites, the $Pb^{2+}$-bromo/iodo MOFs exhibit high moisture stability to achieve efficient $CO_2$ photoreduction by oxidation of water vapor. Both MOFs with the pristine $NH_2$-bdc linkers have a wide visible light absorption, high carrier mobility and μm-ranged long carrier diffusion lengths. Their photocatalytic acitivity in $CO_2$ reduction (AQY of 1.4% at 400 nm, 1.58 wt.% Ru cocatalyst) is the highest reported value in organolead halide hybrids that using water vapor as the sacrificial agent[30]. The performance is also higher than many benchmark metal-oxo MOFs with the identical organic linker (*e.g.* UiO-66(Zr)-$NH_2$ and MIL-101(Fe)-$NH_2$) as well[13,31].

## Results

### Synthesis and structural determination of TMOF-10-NH_2

Solvothermal reactions of $PbX_2$ (X = $Br^-$/$I^-$), $NH_2$-bdc and perchloric acid in DMF/EtOH afforded the brown plate-like crystals of $[Pb_2X]^{3+}(NH_2$-bdc)_2 • G (X = $Br^-$/$I^-$, G = guest: $(CH_3)_2NH_2^+$, TMOF-10-$NH_2$(Br) and TMOF-10-$NH_2$(I), TMOF = Tongji MOF) (Supplementary Fig. 1). Perchloric acid performs as an inorganic acid to regulate the synthetic pH and a crystallization stabilizer, which is analogous to the hydrofluoric acid in zeolite synthesis[32]. To note, the addition of stoichiometric amount of deionized water during synthesis is essential for crystal growth of TMOF-10-$NH_2$ (Supplementary Fig. 2)[33]. Fourier transform infrared (FT-IR) spectra suggest the presence and deprotonation of the $NH_2$-bdc in both TMOF-10-$NH_2$ materials (Supplementary Fig. 3).

X-ray crystallography reveals that the successful isoreticular synthesis of TMOF-10-$NH_2$(Br) and TMOF-10-$NH_2$(I) with 3D coordination networks and 1D pore channels (Fig. 1a). Both structures possess parallel packing of infinite rod-shaped $[Pb_2X]^{3+}$ (X = $Br^-$/$I^-$) SBUs that are bridged by $NH_2$-bdc as struts. The rare bromo/iodo-bridged rod-shaped SBUs consist of corner-sharing $[Pb_2X_2]^{2+}$ (X = $Br^-$/$I^-$) square planar units that propagate along *a*-axis in a zigzag manner (Fig. 1c). This features the $μ_4$-$Br^-$/$I^-$ are the corner-sharing centers and locate in the inner spaces of the rod-shaped SBUs (Fig. 1d). Meanwhile, the outer $Pb^{2+}$ centers of the 8-connected rod-shaped SBUs are coordinated with two $I^-$ and six carboxylate oxygens to afford the distorted dodecahedron coordination geometry (Fig. 1b and Supplementary Fig. 4).

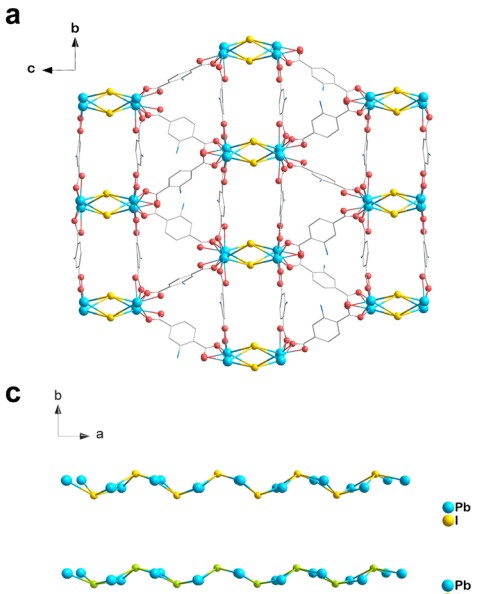

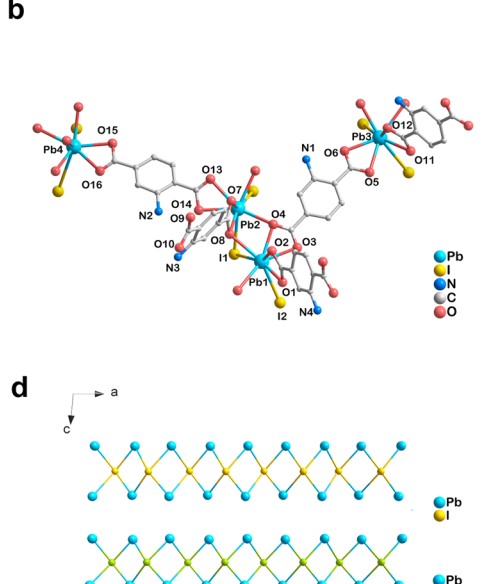

**Fig. 1 | X-ray crystallographic views of TMOF−10-NH₂. a** Crystallographic view of TMOF-10-$NH_2$(I) along the *a*-axis. **b** Crystallographic view of coordination environments for TMOF-10-$NH_2$(I). Crystallographic view of a single $[Pb_2X]^{3+}$ (X = Br, I)

chain in TMOF-10-$NH_2$ along *c*-axis (**c**) and *b*-axis (**d**), respectively. Pb sky blue, I gold, Br green, N blue, O red, C grey. H atoms are omitted for clarity.

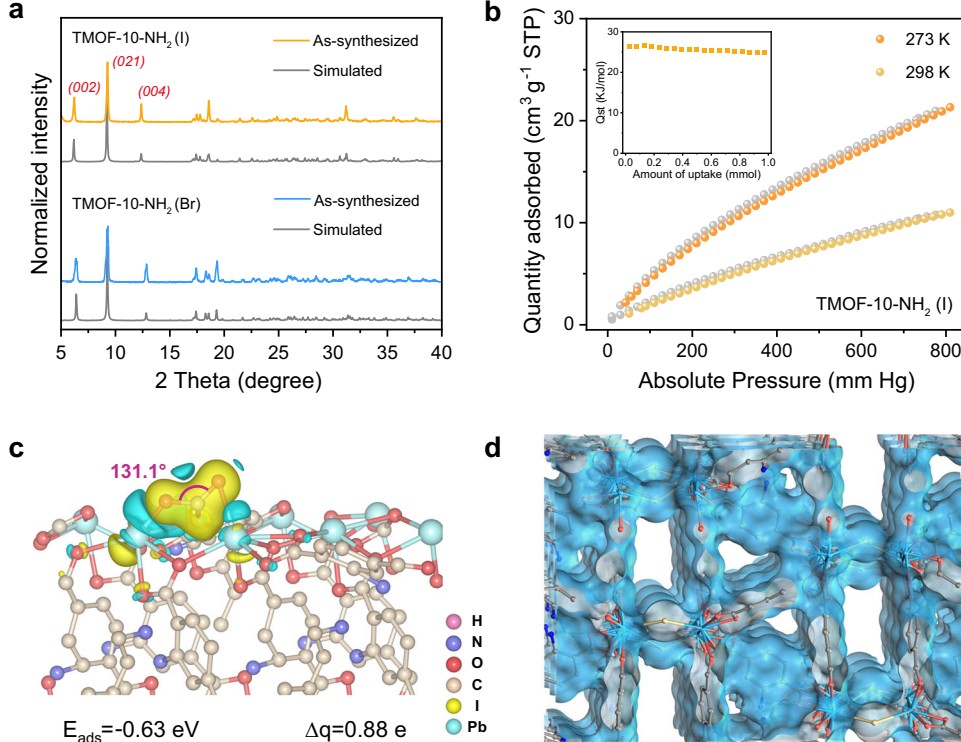

**Fig. 2 | PXRD and CO₂ sorption of TMOF-10-NH₂. a** Experimental and simulated PXRD patterns for TMOF-10-NH₂. **b** CO₂ absorption isotherms of TMOF-10-NH₂(I) at 273 K and 298 K. Inset is the Qst of CO₂ absorption. **c** Differential charge density of the adsorbed CO₂ molecule on TMOF-10-NH₂(I) along (*001*) plane. E_ads adsorption energy, Δq difference of Bader charge. **d** Simulated pore channel diagram of TMOF-10-NH₂(I).

## Stability and CO₂ Sorption of TMOF-10-NH₂

The high yield (~80%, gram-scale synthesis) and high phase purity of both TMOF-10-NH₂(Br) and TMOF-10-NH₂(I) were confirmed by C/H/N elemental analyses and experimental powder X-ray diffraction patterns (PXRD), which matched well with the simulated patterns from the single-crystal data (Fig. 2a). Thermogravimetric analysis and ex-situ thermodiffraction demonstrated high thermal stability up to 200 °C in air (Supplementary Figs. 5–8). Importantly, TMOF-10-NH₂ also exhibited high moisture stability in a wide range of humidity (50–90% relative humidity, RH), which was evidenced by the PXRD patterns after incubation of as-synthesized MOFs under these conditions for 24 h (Supplementary Figs. 9, 10). The substantially enhanced moisture stability over the lead halide perovskites is ascribed to the high-coordinate bromo/iodo atoms residing in the inner spaces of SBUs[29]. The high photostability of TMOF-10-NH₂ was also studied by continuous light irradiation (300 W Xe lamp, 24 W cm⁻²) for 48 h under 90% RH at room temperature (Supplementary Figs. 11, 12). To conclude, the stability tests show that TMOF-10-NH₂ are a robust catalytic platform to perform photoreduction of CO₂ in H₂O vapor.

The open topology affords the MOFs with an array of 1D pore channels along *a*-axis (Fig. 2d). Interestingly, despite the porosity of both TMOF-10-NH₂ (I) and TMOF-10-NH₂(Br) filled with (CH₃)₂NH₂⁺ cations, the pore spaces were successfully activated by incubation in EtOH solvent at elevated temperature, followed by vacuum drying at 80 °C for overnight. To note, the vast majority of (CH₃)₂NH₂⁺ guests were removed during the activation process, suggested by ¹H NMR of digested TMOF-10-NH₂(I) by diluted HF/*d*⁶-DMSO (Supplementary Fig. 13). The leaching of (CH₃)₂NH species into EtOH again confirmed by ¹H NMR of the EtOH supernatant (Supplementary Fig. 14). Meanwhile, the amino groups of TMOF-10-NH₂(I) were found to be partially protonated after the activation process, which is reasonable to compensate the charge balance of MOF. The activated TMOF-10-NH₂(I) exhibits a CO₂ uptake amount of 21.47 cm³/g at 273 K and 11.12 cm³/g at

298 K under 1 bar (Fig. 2b), which afford the zero-coverage isosteric heat of adsorption (Q_st) to be 26.3 kJ mol⁻¹. Meanwhile, TMOF-10-NH₂(Br) shows analogous performance in CO₂ uptake, and the Q_st is determined to be 31.38 kJ mol⁻¹ (Supplementary Fig. 15). The surface areas are measured to be 51.6 m²/g for TMOF-10-NH₂(I) and 53.7 m²/g for TMOF-10-NH₂(Br), respectively, by CO₂ isotherms at 273 K using the BET theory[34]. The low surface area values and confined porosity are further confirmed by theoretical calculations (85.1 m²/g for TMOF-10-NH₂(I) and 85.7 m²/g for TMOF-10-NH₂(Br)) by Materials Studio. The adsorption isotherm of water vapor was also studied to show an H₂O uptake amount of 13.03 cm³/g at 298 K for TMOF-10-NH₂(I) (Supplementary Fig. 16). These values suggest that the porosity of our MOFs is accessible to both CO₂ and H₂O vapor, therefore showing the high potentials for gas-phase catalysis.

Studying the exposed lattice facet is equally important in intrinsic crystalline photocatalysts[35,36]. Herein, the CO₂ adsorption behavior on the different exposed facets of TMOF-10-NH₂(I) were simulated by density functional theory (DFT) calculations. The calculation results show that the adsorption of CO₂ is thermodynamically favorable on the (*001*) facets, which are the major exposed crystal facets of TMOF-10-NH₂(I) (Supplementary Fig. 17). DFT calculations indicate the adsorption energy (E_ads) of CO₂ on TMOF-10-NH₂(I) to be −0.63 eV, which is more negative than many reported metal halide perovskites (Fig. 2c)[37]. This implies a strong CO₂ binding towards the metal halide SBUs in our MOFs. Moreover, the distinct charge accumulations around CO₂ molecules suggest a significant charge transfer of ~0.88 electrons from TMOF-10-NH₂(I) to CO₂, based on the Bader method. The adsorbed CO₂ (denoted as *CO₂) features a bent configuration with O=C=O bond angle of 131.1° and suggests the electron injection into the antibonding 2π_u orbitals[38], confirming the potential activation of CO₂ molecules by SBUs in TMOF-10-NH₂(I). The strong host-guest interaction is probably induced by the strong interactions between the O atoms in *CO₂ and the exposed Pb²⁺ centers in TMOF-10-NH₂(I).

 

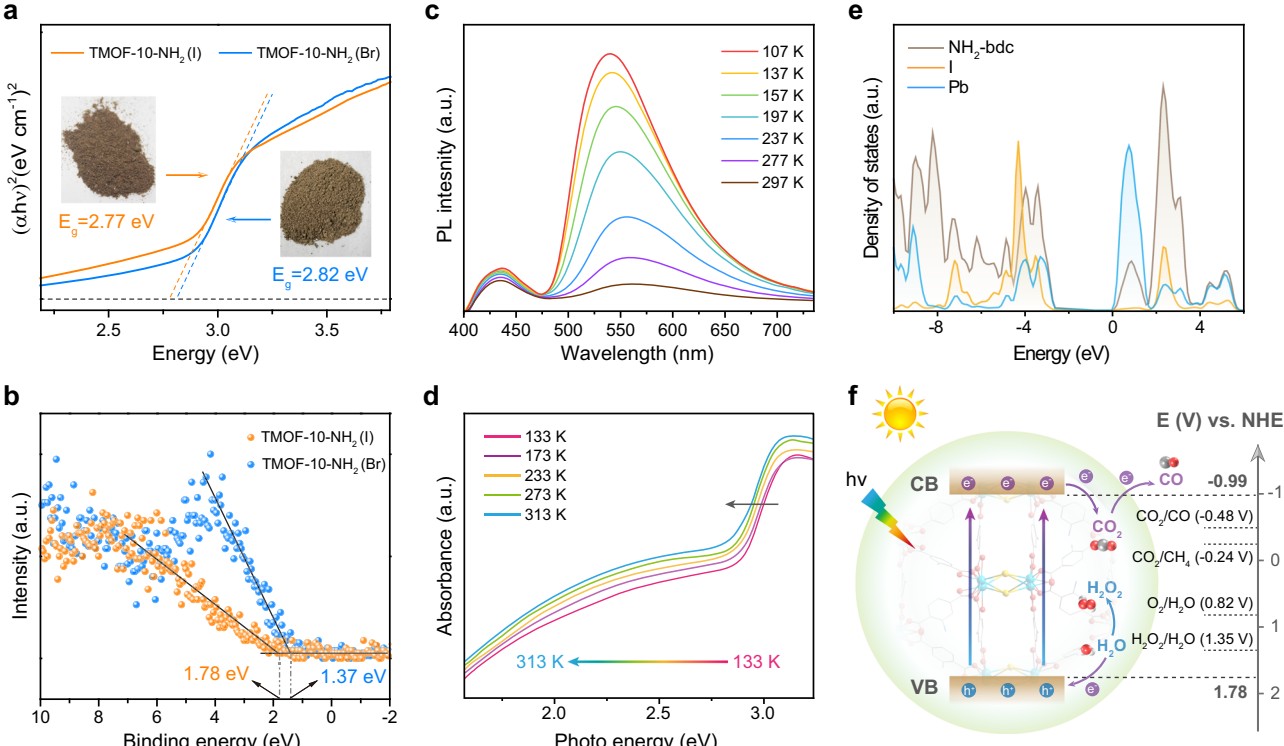

**Fig. 3 | Photophysical properties of TMOF-10-NH₂. a** Estimated band gaps of TMOF-10-NH₂(I) and TMOF-10-NH₂(Br) by UV-Vis absorption spectra. **b** VB-XPS spectra of TMOF-10-NH₂(I) and TMOF-10-NH₂(Br). **c** Temperature-dependent emission spectra of TMOF-10-NH₂(I) from 107 K to 297 K. **d** Temperature-dependent UV-Vis diffuse reflectance spectra of TMOF-10-NH₂(I) from 133 K to 313 K. **e** Calculated DOS for NH₂-bdc (gray), I (yellow) and Pb (blue) in TMOF-10-NH₂(I). **f** Schematic band structure diagram for TMOF-10-NH₂(I). a.u. arbitrary units, CB conduction band, VB valence band, NHE normal hydrogen electrode.

## Band structure of TMOF-10-NH₂

The band structures of TMOF-10-NH₂ were studied by ultraviolet–visible (UV-Vis) diffusion reflectance spectroscopy and valence band X-ray photoelectron spectroscopy (VB-XPS). According to the Kubelka-Munk method, the Tauc plots give the best fit of direct bandgaps for both materials, showing 2.82 eV for TMOF-10-NH₂(Br) and 2.77 eV for TMOF-10-NH₂(I), respectively (Fig. 3a and Supplementary Fig. 18). Moreover, both materials show a low-energy Urbach tail to extend the visible-light absorption to 750 nm. In order to investigate the origin of the Urbach tail, we first performed the variable-temperature UV-Vis absorption spectra of TMOF-10-NH₂(I). The temperature-dependent Urbach tail from 133 K to 313 K (Fig. 3d) suggest that its intrinsic nature which arises from the short-range localization of excitons coupling to lattice distortions[39,40]. The extrinsic crystal defects have negligible contribution, which often provides the temperature-independent Urbach tail. The intrinsic nature was further supported by the good agreement between the calculated and observed values in C/H/N elemental analysis (Supplementary Table 5). In addition, no apparent grain boundaries were found on the surface of the TMOF-10-NH₂(I) single crystals (Supplementary Fig. 19). Moreover, this class of organolead halide materials with asymmetric 1D lead halide chains are widely studied to have populated self-trapped excitons, which originate from strong electron-phonon coupling in the deformable lattice[41,42]. The steady-state photoluminescence (PL) spectra of both TMOF-10-NH₂ materials confirmed the presence of a low-energy broadband emission originating from self-trapped excitons centered at ~540 nm as well as a high-energy band at ~450 nm (Supplementary Figs. 20, 21). The high-energy emission around 450 nm is ascribed to ligand-to-metal charge transfer (LMCT) between the delocalized π-bond of carboxylate groups and $p$ orbitals of Pb²⁺ centers[43,44]. Temperature-dependent steady-state PL studies of TMOF-10-NH₂(I) suggest the narrower and more intense self-trapped

emission peaked at ~540 nm when temperature decreasing from 297 K to 107 K, agreeing with electron-phonon coupling (Fig. 3c). The longitudinal-optical (LO) phonon energy was calculated to be 18(3) meV, according to Eq. 1 (see fitting details and discussions in Supplementary Fig. 22).

$$\Gamma(T) = \Gamma_0 + \Gamma_{LO}(e^{E_{LO}/k_BT} - 1)^{-1} + \Gamma_{inh}e^{-E_b/k_BT} \quad (1)$$

The energy levels of the valence band maximum (VBM) were determined to be 1.78 eV for TMOF-10-NH₂(I) and 1.37 eV for TMOF-10-NH₂(Br), respectively, by VB-XPS (Fig. 3b). The flat-band potentials are evaluated to be −1.11 V vs. Ag/AgCl electrodes (−0.91 V vs. NHE, pH 7) for TMOF-10-NH₂(I) and −1.37 V vs. Ag/AgCl electrodes (−1.17 V vs. NHE, pH 7) for TMOF-10-NH₂(Br) (Supplementary Fig. 23). It is generally known that the CBM of an *n*-type semiconductor is -0.2 V more negative than the flat-band potential below the CBM[45]. Therefore, the CBM of TMOF-10-NH₂(I) and TMOF-10-NH₂(Br) are estimated to be ~ −1.11 V vs. NHE and ~−1.37 V vs. NHE, respectively, agreeing with the VB-XPS studies. Therefore, the CBM of both materials are more negative than the redox potentials of CO/CO₂ (−0.48 V vs NHE, pH 7) and CH₄/CO₂ (−0.24 V vs NHE, pH 7), while the VBM are more positive than the redox potential of O₂/H₂O (+0.82 V vs NHE, pH 7) and H₂O₂/(H₂O) (+1.35 V vs NHE, pH 7) (Fig. 3f and Supplementary Fig. 24). These values confirm that TMOF-10-NH₂ are thermodynamically feasible to achieve both the half-reaction of CO₂ reduction and the half-reaction of H₂O oxidation.

The band structures of TMOF-10-NH₂(I) were further studied by DFT calculations. The calculated bandgap of TMOF-10-NH₂(I) is 2.76 eV (Supplementary Fig. 25), corresponding well with the experimental value (2.77 eV). The total density of states (DOS) and the projected density of states (pDOS) on the Pb, I and C orbitals

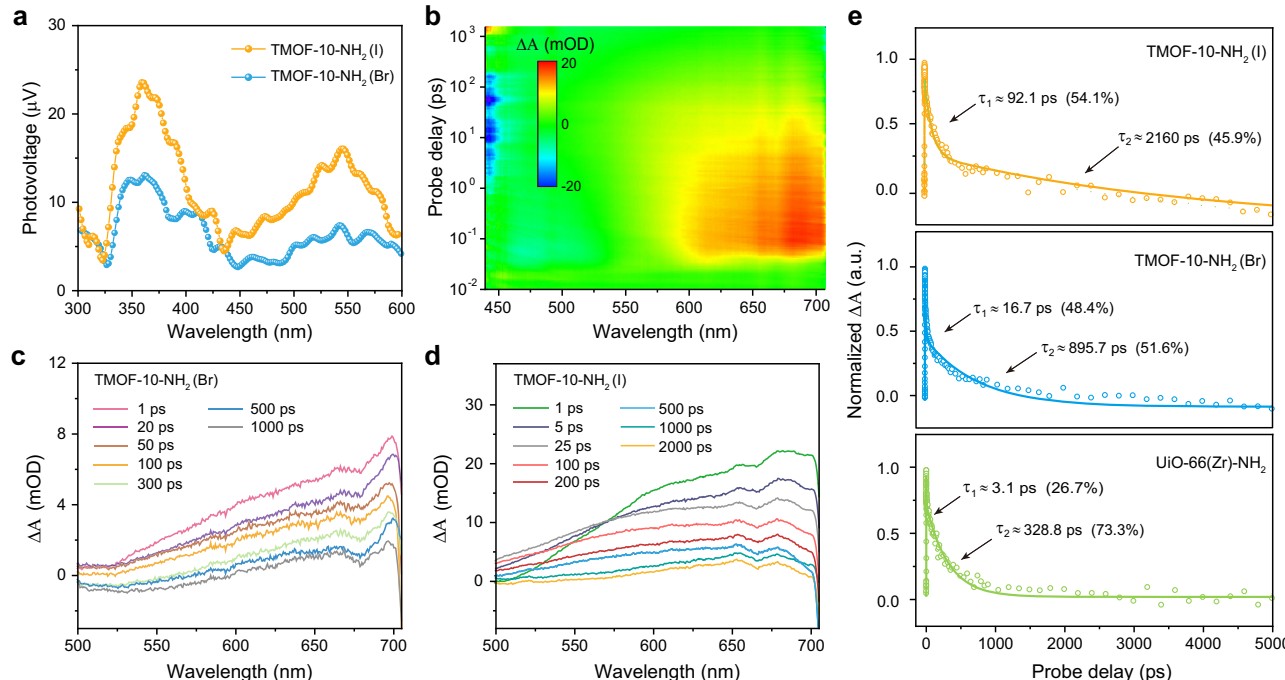

**Fig. 4 | SPV and femtosecond ultrafast TA measurements of TMOF-10-NH₂.**
**a** SPV spectra of TMOF-10-NH₂(I) and TMOF-10-NH₂(Br). **b** Two-dimensional pseudo-color TA plot of TMOF-10-NH₂(I). **c, d** TA spectra of TMOF-10-NH₂(Br) and TMOF-10-NH₂(I) measured at different delay times, respectively. **e** TA kinetics of TMOF-10-NH₂(I), TMOF-10-NH₂(Br) and UiO-66(Zr)-NH₂.

suggest that conduction bands minimum (CBM) is dominated by Pb 6$p$ orbitals, while NH₂-bdc linkers contribute to valence band maximum (VBM) (Fig. 3e and Supplementary Fig. 26). The effective masses ($m_e^*$ and $m_h^*$) of TMOF-10-NH₂(I) are further calculated to be $0.67m_0$ and $0.18m_0$, respectively. The values are comparable to organolead halide perovskites ($m_e^*$: $0.17-0.73m_0$ and $m_h^*$: $0.28-0.36m_0$ for MAPbI₃) and substantially lower than vast majority of benchmark metal-oxo MOFs[46,47]. Based on the DOS calculations as well as the high-energy photoluminescence band at ~450 nm, it is reasonable to attribute this to the charge transfer from NH₂-bdc to Pb²⁺ centers residing in bromo/iodo-bridged SBUs[47,48]. This results in the spatial charge separation upon light irradiation, owing to the high carrier mobility and long carrier diffusion length in bromo/iodo-bridged SBUs.

**Carrier transport properties of TMOF-10-NH₂**
The intrinsic carrier-transport characteristics of photocatalysts are equally important features to the photocatalysts. The traditional metal-oxygen clusters in MOFs often suffer from the low electrical conductivity and poor carrier mobility, *e.g.* UiO-66-NH₂, therefore exhibiting short carrier lifetime and fast electron-hole recombination[49]. However, the distinctive electronic configuration of Pb²⁺ with strong spin-orbit coupling that are bridged by soft Br⁻/I⁻ anions endow the lead halide units with small effective mass and high mobility[46].

First, the surface photovoltage spectroscopy (SPV) was employed to study the photoinduced electron-hole separation and carrier transport behaviors. Two irradiation-induced photovoltage signal peaks at ~370 nm and ~550 nm were observed for both TMOF-10-NH₂(I) and TMOF-10-NH₂(Br) (Fig. 4a). Based on the band structure of both TMOFs, the PV response at ~370 nm is determined to be the electronic transition from the valence band to the conduction band. Meanwhile, the broadband at ~550 nm indicates the transition from the valence band to the surface localized electronic states and/or from the surface localized electronic states to the conduction band[50]. Upon light irradiation, both the extended (band states) and localized (tail states) photocarriers are effectively separated. The higher SPV responses of

TMOF-10-NH₂(I) over TMOF-10-NH₂(Br) indicate a more favorable carrier transport in TMOF-10-NH₂(I), agreeing with many studies in organolead halide perovskites[51,52].

To gain deeper insight of carrier-transport characteristics, ultrafast transient absorption (TA) spectroscopy (the pump laser is set as 385 nm, above the band gap, the range of probe wavelength is 430–770 nm) was employed to study the real-time photoexcited carrier dynamics of both TMOF-10-NH₂. Figure 4b shows a pseudo-color representation of TA spectrum as functions of probe wavelength and pump–probe delay times for TMOF-10-NH₂(I). Corresponding to the contour map, a subset of representative TA spectra taken at different probe delays after excitation show broad positive absorption features peaking from 525 to 705 nm for both TMOF-10-NH₂(Br) and TMOF-10-NH₂(I) (Fig. 4c, d). The analogous transient absorption trend implies the similar relaxation processes for TMOF-10-NH₂(Br) and TMOF-10-NH₂(I) upon above band-gap excitation. The analysis of the recovery kinetics shows that the best fit of the decay provides a biexponential function with two time constants of $\tau_1 = 91.1$ ps, $\tau_2 = 2160$ ps for TMOF-10-NH₂(I) and $\tau_1 = 16.7$ ps, $\tau_2 = 895.7$ ps for TMOF-10-NH₂(Br), respectively (Fig. 4e). The $\tau_1$ and $\tau_2$ are ascribed to the electron dynamics associated with the different electron trap states that are energetically located within the bandgap of TMOF-10-NH₂[53,54]. These two near-band-edge trap states accumulate the photogenerated electrons from the bottom of conduction band in a bi-exponential relaxation manner[55]. The lifetimes of such long-lived trap states are typically in the nanosecond domain, therefore the electron-detrapping processes are further examined by time-resolved photoluminescence spectroscopy that will be discussed later (Supplementary Figs. 28, 29)[56]. In order to directly compare the carrier transport characteristics, the TA study of a benchmark metal-oxo MOF with the identical organic linker, i.e. UiO-66(Zr)-NH₂, was performed under the same condition for comparison. A positive absorption peak from 550 nm to 700 nm was observed and the recovery kinetics was fitted with time constants of $\tau_1 = 3.1$ ps and $\tau_2 = 328.8$ ps, substantially shorter than both TMOF-10-NH₂ (Fig. 4e). Moreover, the stronger positive absorption of

**Table 1 | Carrier transport characteristics of TMOF-10-NH$_2$ and UiO-66(Zr)-NH$_2$**

| Materials | Band gap [eV] | Mobility [cm$^2$ V$^{-1}$ s$^{-1}$] | Carrier concentration [cm$^{-3}$] | TA lifetimes [ps] | PL lifetimes [ns] | Diffusion length [μm] |
|---|---|---|---|---|---|---|
| TMOF–10-NH$_2$(I) | 2.77 | 3.07 | 1.28×10$^{15}$ | τ$_1$ = 91.1 τ$_2$ = 2160 | τ$_1$ = 2.28 τ$_2$ = 26.66 | 0.13–0.45 |
| TMOF-10-NH$_2$(Br) | 2.83 | 0.53 | 8.87×10$^{14}$ | τ$_1$ = 16.7 τ$_2$ = 895.7 | τ$_1$ = 1.84 τ$_2$ = 21.39 | 0.05–0.17 |
| UiO-66(Zr)-NH$_2$ | 2.76 | n.d. | n.d. | τ$_1$ = 3.1 τ$_2$= 328.8 | n.d. | n.d. |

*n.d. not determined

TMOF-10-NH$_2$ over than UiO-66(Zr)-NH$_2$ suggests higher concentration of photogenerated electrons in the bromo/iodo-bridged MOFs (Supplementary Fig. 27).

The longer carrier lifetime and stronger positive absorption of TMOF-10-NH$_2$(I) are further evidenced by the Hall effect measurement. The Hall effect measurement at room temperature indicated that *n*-type semiconductive nature of TMOF-10-NH$_2$(I) with a carrier concentration of 1.28 × 10$^{15}$ cm$^{-3}$ and an estimated carrier mobility of 3.07 cm$^2$ V$^{-1}$ s$^{-1}$. Meanwhile, TMOF-10-NH$_2$(Br) has a carrier concentration of 8.87 × 10$^{14}$ cm$^{-3}$ and a carrier mobility of 0.53 cm$^2$ V$^{-1}$ s$^{-1}$, respectively. Time-resolved photoluminescence decay studies were further employed to measure the carrier lifetimes of TMOF-10-NH$_2$(I) and TMOF-10-NH$_2$(Br) (Supplementary Fig. 28). Both decay curves were fitted using a two-component exponential method[57], demonstrating that TMOF-10-NH$_2$(I) exhibits a short lifetime of 2.28 ± 0.04 ns and a long decay time of 26.66 ± 0.32 ns. Both values are longer than the counterparts of TMOF-10-NH$_2$(Br) (τ$_1$ = 1.84 ± 0.03 ns and τ$_2$ = 21.39 ± 0.26 ns). The carrier diffusion lengths $L_D$ were estimated to be in the range of 0.13–0.45 μm for TMOF-10-NH$_2$(I), and 0.05–0.17 μm for TMOF-10-NH$_2$(Br), respectively, based on the the equation:

$$L_D = (k_B T / e \times \mu \times \tau)^{1/2} \qquad (2)$$

where $k_B$ is the Boltzmann's constant, $T$ is the absolute temperature, $\mu$ is the carrier mobility, and $\tau$ is the carrier lifetime (Table 1)[58]. Overall, the high carrier mobility and long carrier diffusion length in TMOF-10-NH$_2$(I) were thoroughly studied by a variety of photophysical studies, including SPV, TA and Hall effect measurements.

In order to further investigate the critical contribution of halide species, photophysical studies of a Pb$^{2+}$-based MOF, [Pb(NH$_2$-bdc)]$_n$, have been performed. First, X-ray crystallography of [Pb(NH$_2$-bdc)]$_n$ reveals that its 3D coordination network has 1D pore channels analogous to TMOF-10-NH$_2$, but occupying 1D Pb$^{2+}$-carboxylate chains without the presence of halide species (Supplementary Figs. 30, 31). UV-Vis spectroscopy shows a bandgap of 2.43 eV for [Pb(NH$_2$-bdc)]$_n$ close to the bandgaps of TMOF-10-NH$_2$ (Supplementary Fig. 32), but its carrier transport characteristics are inferior to TMOF-10-NH$_2$. SPV spectroscopy indicates a very weak photovoltage response at -370 nm for [Pb(NH$_2$-bdc)]$_n$, suggesting the unfavorable carrier diffusion process (Supplementary Fig. 33). In addition, the transient-state photoluminescence spectroscopy shows a fast decay lifetime of 0.35 ns for [Pb(NH$_2$-bdc)]$_n$, substantially lower than average lifetimes of TMOF-10-NH$_2$(Br/I) (11.83–14.78 ns) (Supplementary Figs. 34, 35). Overall, the photophysical studies of the control material, [Pb(NH$_2$-bdc)]$_n$, unambiguously evidence the important role of bridging halide species of TMOF-10-NH$_2$ in enhancing their intrinsic carrier transport.

## Overall photocatalytic CO$_2$ reduction and H$_2$O oxidation

The photocatalytic CO$_2$ reduction was initially studied by introducing 10 mg as-synthesized TMOF-10-NH$_2$ into a sealed reaction system which contained 5 mL H$_2$O with CO$_2$ pressure of 1 atm without any co-catalysts or sacrificial reagents, the TMOF-10-NH$_2$ samples were uniformly dispersed on a quartz filter membrane in the center of the reaction cell, which avoids the direct contact with H$_2$O. Such physical setup along with the gas-phase catalytic reaction effective overcome the Pb$^{2+}$ leaching problem from the lead halide hybrids, and negligible amount of Pb$^{2+}$ leaching was detected in the aqueous solution

throughout the photocatalysis (<0.1 ppm by inductively-coupled plasma optical emission spectroscopy, ICP-OES). The gas chromatography (GC) was employed to identify and quantify the gas products (Supplementary Fig. 36), and the liquid oxidation production H$_2$O$_2$ was quantitatively determined by colorimetry test (discussed later). Upon the AM1.5 G simulated illumination, TMOF-10-NH$_2$ steadily photocatalyzed CO$_2$ reduction to CO as the major product over a span of 4 h, and only trace amount of CH$_4$ were observed (Fig. 5a). To note, no H$_2$ and O$_2$ product were observed in our gas-phase catalytic system. The average CO evolution rates were determined to be 78 μmol h$^{-1}$ g$^{-1}$ for TMOF-10-NH$_2$(I) and 52 μmol h$^{-1}$ g$^{-1}$ for TMOF-10-NH$_2$(Br), respectively, superior to the vast majority of MOFs in photoreduction of CO$_2$ with water vapor (Supplementary Table 3). The higher photocatalytic performance of iodide-base MOF over the bromide analog is ascribed to the better carrier transport in the TMOF-10-NH$_2$(I), which has been evidenced by the higher SPV responses and longer lifetime constants in TA spectroscopy studies (Fig. 4a–e). The photocatalytic stability of TMOF-10-NH$_2$(I) was studied by using the same catalyst over three continuous cycles by evacuation and filling with CO$_2$, and the photocatalytic activity in the second and the third run was largely maintained (Fig. 5b). Moreover, no obvious decrease of the CO evolution rate was observed after the continuous photocatalytic reaction for 24 h (Fig. 5c). The post-photocatalysis PXRD, FT-IR and SEM studies show negligible change in the as-prepared sample, further confirming the high stability and excellent durability of TMOF-10-NH$_2$(I) towards CO$_2$ photoreduction (Supplementary Figs. 37–39).

To verify the origin of carbon production, the control experiments and the $^{13}$C isotopic labeling experiment were performed. All of the control photocatalytic experiments without photocatalysts or in the dark produced negligible amount of CO, CH$_4$ and H$_2$O$_2$, showing the photo-driven CO$_2$-based catalytic conversion (Supplementary Figs. 40, 41). For the $^{13}$CO$_2$ labeling experiment, the mass spectroscopy (MS) with the signal peaks at m/z = 29 ($^{13}$CO) was observed, implying the generated carbon products originating from the CO$_2$ reduction rather than other carbon sources (Fig. 5d).

Meanwhile, the half reaction of water oxidation was also studied to produce H$_2$O$_2$ during the photocatalytic process (confirmed by commercial colorimetric test strips and blank control experiment, Supplementary Figs. 41, 42), demonstrating the coupling of H$_2$O oxidation and CO$_2$ reduction to realize the artificial photosynthesis. The amount of H$_2$O$_2$ was quantified and confirmed by both colorimetry test[59], and the N,N-diethyl-1,4-phenylene-diamine-peroxidase (DPD/POD) methods[60]. A linear increment in the H$_2$O$_2$ concentrations was observed in a 12 h run and the average H$_2$O$_2$ evolution rate was determined to be 36.3 μmol h$^{-1}$ g$^{-1}$ (Fig. 5e and Supplementary Figs. 43–44), which is -0.92 times of the CO generation rate. To note, the slight difference of the hole and electron consumption ratio is probably due to the formation of other undetected reactive oxygen species (ROS)[36]. In addition, XPS studies of post-catalysis TMOF-10-NH$_2$(I) have been performed to exclude the possible self-oxidation of photocatalysts (Supplementary Fig. 45).

To verify the photocatalytic performance of TMOF-10-NH$_2$, the benchmark MOFs with photocatalytically active metal-oxo SBUs (i.e. MIL-101(Fe)-NH$_2$[61], UiO-66(Zr)-NH$_2$[48], [Pb(NH$_2$-bdc)]$_n$ and 1D organo-lead iodide perovskite (C$_6$H$_{10}$N$_2$)[PbI$_4$][62] were synthesized and studied in photocatalytic CO$_2$ reduction under the identical reaction conditions (Supplementary Figs. 46–48). Despite the high stability of MOFs,

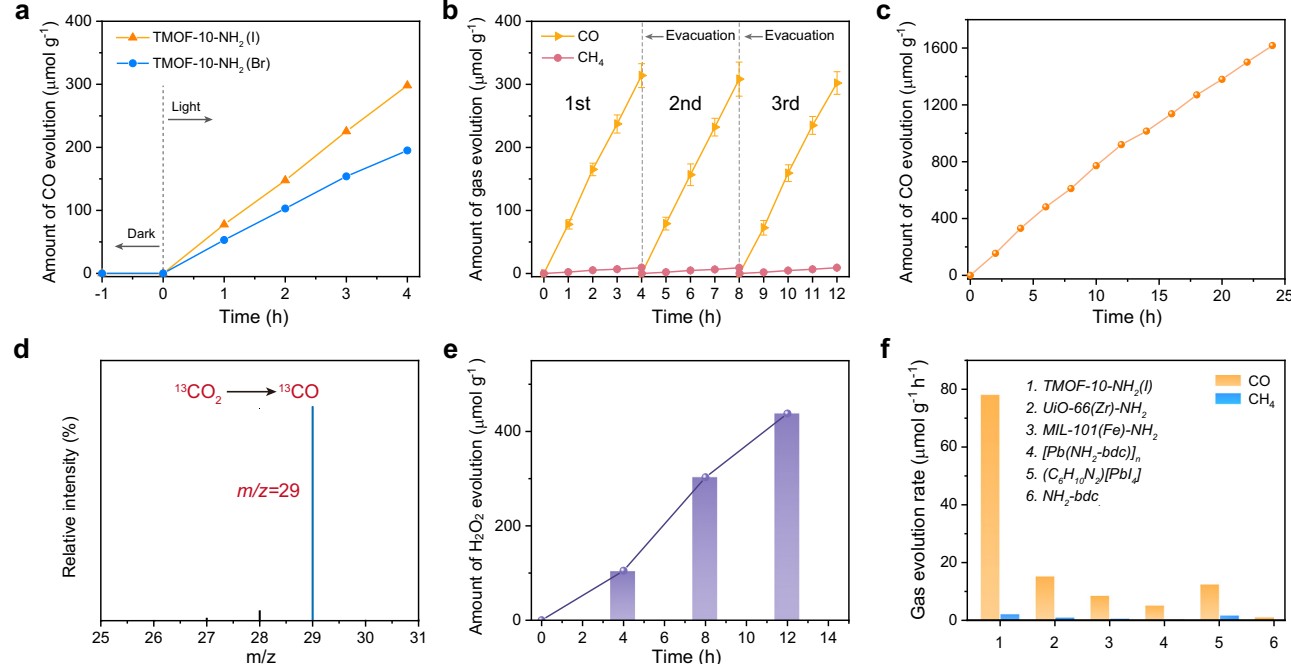

**Fig. 5 | Overall photocatalytic $CO_2$ reduction performance of TMOF–10-NH₂.**
**a** Time courses of CO evolution by photocatalytic $CO_2$ reduction with water vapor using TMOF-10-NH₂ as photocatalysts under AM1.5 G simulated sunlight. **b** Time courses of photocatalytic $CO_2$ reduction using TMOF-10-NH₂(I) for 12 h, with evacuation every 4 h. Error bars represent the standard deviations of photocatalytic performance based on three independent samples. **c** Long-term photoreduction $CO_2$ of TMOF-10-NH₂(I). The system was continuous irradiated under AM1.5 G simulated sunlight for 24 h and gas products were measured every 2 hours. **d** GC-MS results of $^{13}CO$ produced over TMOF-10-NH₂(I) from $^{13}CO_2$ isotope experiment in water vapor. **e** Amount of $H_2O_2$ produced as a function of the time. **f** Comparison of gas evolution rates between TMOF-10-NH₂(I), UiO-66(Zr)-NH₂[48], MIL-101(Fe)-NH₂[61], [Pb(NH₂-bdc)]ₙ, 1D (C₆H₁₀N₂)[PbI₄] perovskite[62] and NH₂-bdc ligands.

the CO evolution rates afford to be 8.6 μmol h⁻¹ g⁻¹ for MIL-101(Fe)-NH₂, 15.3 μmol h⁻¹ g⁻¹ for UiO-66(Zr)-NH₂, and 5.2 μmol h⁻¹ g⁻¹ for [Pb(NH₂-bdc)]ₙ, substantially lower than TMOF-10-NH₂(I) (Fig. 5f). The 1D organolead iodide perovskite exhibits the product evolution rates of 1.8 μmol h⁻¹ g⁻¹ for CH₄ and 12.6 μmol h⁻¹ g⁻¹ for CO, respectively. However, the structure was largely decomposed after one photocatalytic cycle, owing to the moisture-sensitive nature of lead perovskites (Supplementary Fig. 50). These results suggest the photocatalytic performances of our MOFs with metal-iodo SBUs outperforms conventional benchmark metal-oxo MOFs as well as organolead halide perovskites.

## Photocatalytic mechanism

In situ diffuse reflectance infrared Fourier transform spectroscopy (DRIFTS) measurements and DFT calculations were performed to understand the $CO_2$ photoreduction process occurring on TMOF-10-NH₂(I). Upon the irradiation time increasing from 0 to 180 min, two prominent bands ascribed to *COOH groups at 1558 cm⁻¹ and 1637 cm⁻¹ were observed (Fig. 6a)[63,64]. The concomitant increasing intensity implied that the *COOH groups are the intermediates during the photoreduction of $CO_2$ to CO. Moreover, the appearance of the absorption bands at 1420 cm⁻¹ are characteristic of the symmetric stretching of *HCO₃⁻, indicating that $CO_2$ and $H_2O$ molecules were co-adsorbed onto the TMOF-10-NH₂(I)[65,66]. The absorption bands at 1524 cm⁻¹ and 1371 cm⁻¹ are attributed to the formation of monodentate carbonate (m-CO₃²⁻) groups[12], and the bands located at 1331 cm⁻¹ and 1623 cm⁻¹ are assigned to the bidentate carbonate (b-CO₃²⁻) groups[67,68]. The emerging peak at 1251 cm⁻¹ is attributed to the carboxylate (*CO₂⁻) vibration, which facilitates the formation of *COOH[12]. The observed intermediate species confirm the efficient photocatalysis of $CO_2$ reduction. The broad absorption bands at 3250 − 3600 cm⁻¹ are ascribed to the hydroxyl stretching from the adsorbed water (Supplementary Fig. 51)[4]. The absorption peaks at

1280 cm⁻¹ and 2810 cm⁻¹ are assigned to the O−H deformation vibration and the O−H stretching vibration of $H_2O_2$, respectively[69].

Therefore, a rational $CO_2$ photoreduction mechanistic pathway catalyzed by TMOF-10-NH₂(I) is proposed: (1) $CO_2$ and $H_2O$ are initially adsorbed on the catalyst surface. (2) Subsequently, the adsorbed *$CO_2$ molecules interact with the surface protons to form the *COOH intermediate upon illumination. (3) The deprotonation of *COOH intermediate gives the *CO molecules, while the proton transfer affords the absorbed $H_2O$ on MOFs into $H_2O_2$. Based on the spectroscopy investigations, the possible reaction pathway of overall solar $CO_2$ reduction could be proposed as follows:

$$CO_2(g) \rightarrow CO_2* \quad (3)$$

$$*CO_2 + e^- \rightarrow *CO_2^- \quad (4)$$

$$*CO_2^- + H^+ \rightarrow *COOH \quad (5)$$

$$*COOH + H^+ + e^- \rightarrow *CO + H_2O \quad (6)$$

$$*CO \rightarrow CO \quad (7)$$

$$2H_2O \rightarrow H_2O_2 + 2H^+ + 2e^- \quad (8)$$

To confirm the results, we have performed the Gibbs free energy calculations on the possible reaction pathways. The calculated Gibbs free energy of *$CO_2$ formation is lower than the initial values, implying that the $CO_2$ adsorption and activation on TMOF-10-NH₂(I) is energy-favorable (Fig. 6b). The previous studies suggest that the formation of *COOH is the rate-determining step for reduction of $CO_2$ to CO[70]. A low

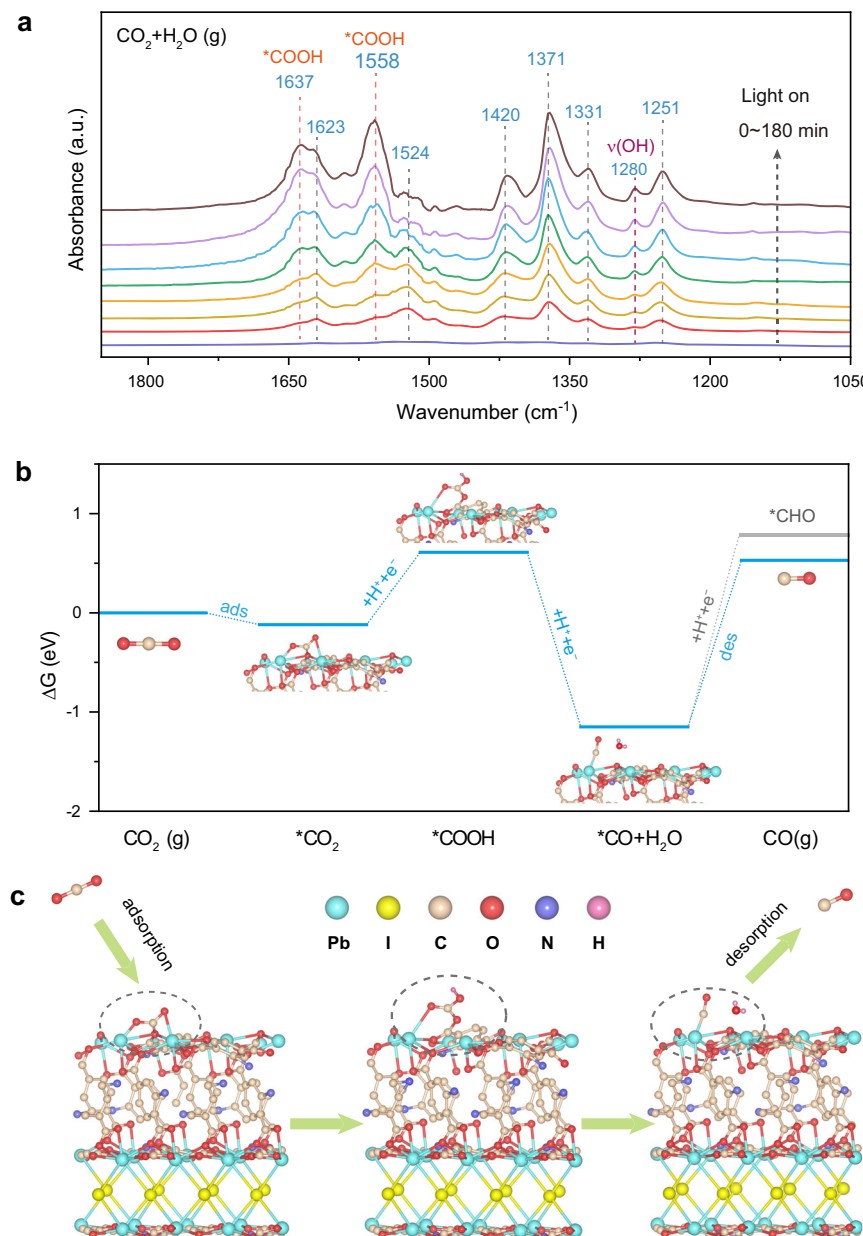

**Fig. 6 | In situ DRIFTS measurement and Gibbs free energy calculations. a** In situ DRIFTS spectra for co-adsorption of a mixture of $CO_2$ and $H_2O$ on the TMOF-10-$NH_2$(I). **b** Calculated free energy ($\Delta G$) diagram of $CO_2$ photoreduction to CO for TMOF-10-$NH_2$(I). **c** Schematic representation of $CO_2$ photoreduction mechanism occurring on TMOF-10-$NH_2$(I).

energy barrier of 0.73 eV was observed for the conversion of $*CO_2$ to $*COOH$ on TMOF-10-$NH_2$(I), which is substantially lower than many previously reported MOF-based catalysts[71,72]. This could be attributed to a more stable binding configuration between COOH* and the surface $Pb^{2+}$ sites. These calculations suggest that TMOF-10-$NH_2$(I) effectively stabilizes the intermediates $*COOH$ for reduction of $CO_2$ to CO, which agrees with aforementioned in situ DRIFTS studies. Moreover, the downhill free energy profiles of protonation $*COOH$ to $*CO$ indicate the spontaneous transformation on the (*001*) plane of TMOF-10-$NH_2$(I), followed by CO release from the dissociation of the weakly bonded $*CO$ adduct as the most favorable product.

Moreover, the high selectivity towards CO generation was further investigated by the Gibbs free energy calculations for CO hydrogenation (Supplementary Fig. 52). The energy for $*CHO$ formation ($\Delta G(*CHO)$) is higher than the desorption energy of CO molecules. This implies that the TMOF-10-$NH_2$(I) is more beneficial for $*CO$ desorption from their surfaces than for the protonation of $*CO$ to produce $*CHO$, which accounts for their nearly quantitative selectivity for visible-light-driven $CO_2$ reduction to CO.

## Deposition of Ru cocatalysts onto TMOF-10-$NH_2$(I)

To further enhance the photocatalytic conversion efficiency, herein, ultrasmall noble metal nanoparticles (NPs) that are widely known to perform as cocatalysts and suppress the electron-hole recombination were attempted to deposit on the crystal surface[73,74]. As shown in the Fig. 7b, the Ru NPs were successful loading into TMOF-10-$NH_2$(I) with the uniform size distribution of 1.5 nm around MOF surface, evidenced by transmission electron microscopy (TEM) images. High-resolution TEM indicates a *d*-spacing of 0.214 nm, corresponding to the (*002*) lattice plane of the hexagonal Ru crystal structure[75]. The excellent dispersion of ultrasmall Ru NPs in the MOF greatly enhances the interface interactions between semiconductors and

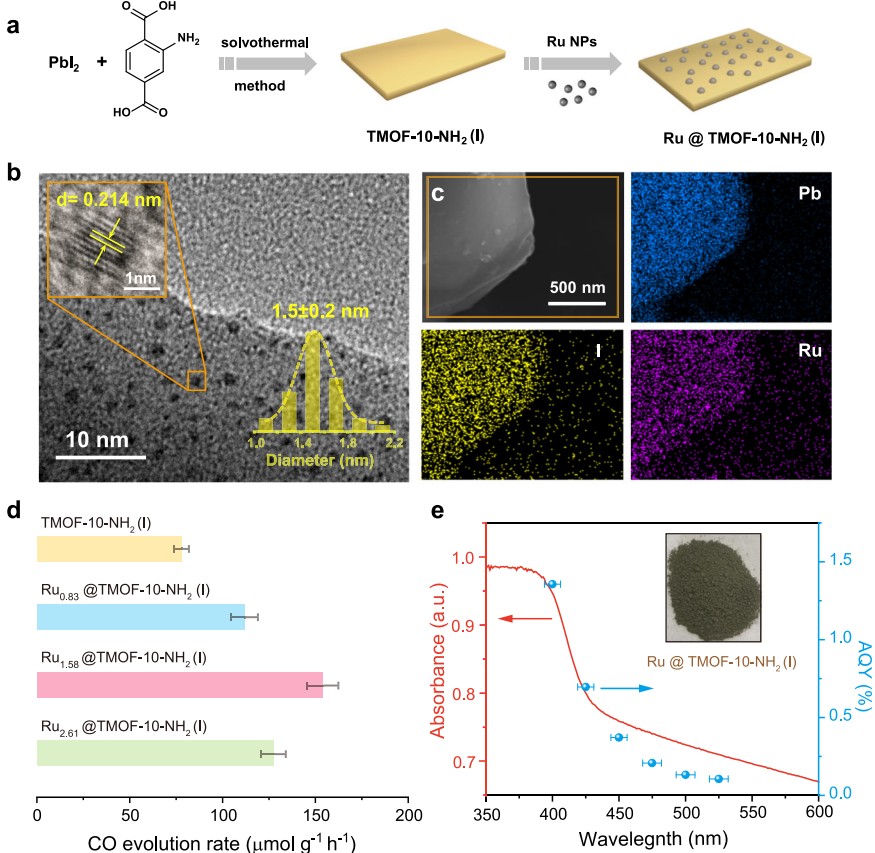

**Fig. 7 | Synthesis and CO₂ photoreduction performance of Ru@TMOF-10-NH₂(I). a** Schematic presentation for synthesis of Ru@TMOF-10-NH₂(I). **b** High resolution TEM images of Ru@TMOF-10-NH₂(I), and the inset shows the size distribution of Ru NPs and lattice spacing of 0.214 nm for Ru NPs. **c** EDS mapping of Ru@TMOF-10-NH₂(I). **d** CO evolution rate of TMOF-10-NH₂(I), Ru₀.₈₃@TMOF-10-NH₂(I), Ru₁.₅₈@TMOF-10-NH₂(I) and Ru₂.₆₁@TMOF-10-NH₂(I). Error bars represent the standard deviations of three independent photocatalysis tests. **e** Wavelength-dependent AQY of CO₂ reduction to CO on Ru@TMOF-10-NH₂(I). Error bars represent the deviations of monochromatic light wavelengths.

metal cocatalysts[76,77]. Elemental mapping of energy-dispersive X-ray spectroscopy (EDS) evidences the presence of Pb, I and Ru in a single crystal of TMOF-10-NH₂(I) (Fig. 7c). The optimized loading amount of Ru NPs in TMOF-10-NH₂(I) was determined to be 1.58 wt.% by ICP-OES. Importantly, a nearly 2-fold enhancement in photocatalytic CO₂ reduction was observed with CO evolution rate of 154 µmol g⁻¹ h⁻¹ (Fig. 7d), exceeding the vast majority of the MOFs and/or lead halide hybrids-based catalysts by using pure water as the sacrificial agent (Supplementary Tables 3, 4). Moreover, no metal leaching or morphology change was observed during the photocatalytic reaction, evidenced by ICP-OES and TEM (Supplementary Fig. 54). The apparent quantum yield (AQY) for Ru@TMOF-10-NH₂(I) at 400 nm was determined to be ~1.36% (Fig. 7e). The AQY decreases with the increase in the irradiated wavelengths, which is consistent with the UV-Vis absorption spectra of Ru@TMOF-10-NH₂(I). This suggests that the CO evolution of the Ru@TMOF-10-NH₂(I) is primarily driven by the photocarriers and strongly dependent on the wavelength of the incident light.

In order to rationalize the improved photocatalytic activity, we first performed XRD and UV-Vis diffusion reflectance spectroscopy to confirm the retention of high crystallinity and the parent band positions after loading Ru NPs (Supplementary Figs. 55, 56). The partial quenching of the photoluminescence and longer average lifetimes consistently indicate that the introduction of Ru NPs effectively suppressed the recombination of radiative electron-hole pairs in TMOF-10-NH₂(I) (Supplementary Figs. 57, 58). Meanwhile, the Ru@TMOF-10-NH₂(I) displays 1.6 times enhancement in the transient photocurrent responses over the pristine TMOF-10-NH₂(I) (Supplementary Fig. 59), and the loading of Ru nanoparticles affords a smaller Nyquist plot diameter in electrochemical impedance spectroscopy (EIS) studies (Supplementary Fig. 60). These studies suggest the substantially enhanced charge separation and transport after incorporation of Ru NPs into our MOFs.

## Discussion

This study shows the synthesis of robust and porous bromo/iodo-bridged SBUs in MOFs for overall photocatalytic CO₂ reduction by water vapor as the electron donor. The materials have both MOF-based porosity for gas-phase reaction and lead perovskitoid-based photo-carrier-transport properties, therefore exhibiting excellent photo-catalytic activity in CO₂ reduction. To the best of our knowledge, the photocatalytic activity for Ru@TMOF-10-NH₂ is higher than all of the previous reported lead halide perovskites as well as many amino-functionalized pristine metal-oxo MOFs. The adsorbed CO₂ structural models along with in situ spectroscopy studies enable our understanding of the photocatalytic mechanism occurring on the lead-iodo SBUs. Moreover, due to the mild gas-phase reaction and no physical contact between aqueous medium and photocatalyst, no Pb²⁺ leaching problem was noticed to cause environmental problems. This study paves the way for MOFs with halide-bridged SBUs as a potentially general class of photocatalysts for small molecule activation and efficient photocarrier transport.

## Methods
### Materials
Lead iodide (PbI₂, 99.9%, Aladdin), 2-aminoterephthalic acid (NH₂-bdc, >98%, Adamas), *N,N*-dimethylformamide (DMF, 99.9%, Aladdin),

ethanol (EtOH, 99.8%, Aladdin) and perchloric acid (70% in $H_2O$, SCR) were used as-received for the solvothermal synthesis of TMOF-10-$NH_2$(I). Lead bromide ($PbBr_2$, 99.9%, Aladdin) were used as-received for the synthesis of TMOF-10-$NH_2$(Br). Lead nitrate ($Pb(NO_3)_2$, >99%, Sinopharm Chemical Reagent Co., Ltd) were used as-received for synthesis of [Pb($NH_2$-bdc)]$_n$. Triruthenium dodecacarbonyl ($Ru_3(CO)_{12}$, 99.95%, Aladdin), tri-$n$-octylamine (TOA, 95%, Aladdin) were used as-received for the synthesis of Ru NPs. All chemicals were obtained from commercial sources and used without further purification.

### Synthesis of TMOF-10-$NH_2$(I) ([$Pb_2I^{3+}$][$^-O_2C(C_6NH_5)CO_2^-$]$_2 \bullet$[$(CH_3)_2NH_2^+$])

A mixture of 0.2305 g $PbI_2$ (0.5 mmol), 0.2715 g 2-aminoterephthalic acid (1.50 mmol), 200 μL perchloric acid ($HClO_4$, 2.42 mmol), and 4 mL mixed solvent of DMF and EtOH ($V_{DMF}$: $V_{EtOH}$ = 2: 1) were added into a 10 mL Teflon-lined autoclave reactor. The autoclave was then sealed into a stainless steel vessel and heated at 150 °C for 72 h. After cooling to room temperature, brownish plate-like crystals of TMOF-10-$NH_2$(I) were isolated after vacuum filtration, and rinsed with EtOH. Yield: 0.182 g (77 % based on total Pb content). The μm-sized microscopic powders of TMOF-10-$NH_2$(I) were prepared via manual grinding.

### Synthesis of TMOF-10-$NH_2$(Br) ([$Pb_2Br^{3+}$][$^-O_2C(C_6NH_5)CO_2^-$]$_2 \bullet$[$(CH_3)_2NH_2^+$])

Crystals of TMOF-10-$NH_2$(Br) can be synthesized in the same manner as for TMOF-10-$NH_2$(I) but with $PbBr_2$ in place of $PbI_2$ at the same molar ratio. Yield: 0.175 g (78 % based on total Pb content).

### Synthesis of [Pb($NH_2$-bdc)]$_n$

A mixture of 0.083 g $Pb(NO_3)_2$ (0.25 mmol), 0.042 g 2-aminoterephthalic acid (0.25 mmol) and 2 mL DMF were added into a 10 mL Teflon-lined autoclave reactor. The autoclave was then sealed into a stainless steel vessel and heated at 120 °C for 48 h. After cooling to room temperature, yellow crystals of [Pb($NH_2$-bdc)]$_n$ were isolated after vacuum filtration, and rinsed with EtOH. Yield: 0.078 g (81 % based on total Pb content).

### Synthesis of ultrasmall Ru NPs

The -1.5 nm Ru NPs were synthesized by thermal decomposition of $Ru_3(CO)_{12}$[74]. Specifically, 8 mL of TOA and 25 mg of $Ru_3(CO)_{12}$ (0.039 mmol) were added to into a 25 mL three-neck round bottom flask. The reaction mixture was degassed for 30 min at room temperature. The flask was then flushed with nitrogen, heated to 270 °C and kept at this temperature for 30 min. The particles were washed with EtOH twice and isolated by centrifugation (8500 rpm for 3 min), and finally dispersed in toluene, affording black colloidal solution of Ru NPs with a concentration of approximately 1 mg/mL (Supplementary Fig. 53).

### Synthesis of Ru@TMOF-10-$NH_2$(I)

In a typical synthesis, 60 mg of as-prepared TMOF-10-$NH_2$(I) was fully dispersed in 10 mL DMF solution, and 0.2–1 mL Ru NPs solution (Ru: DMF = 2.0, 4.8, and 9.1 v/v%) was added under vigorous stirring, followed by sonication for 30 min. Then, the mixture was incubated at room temperature for 12 h. The gray precipitates were separated by centrifugation (5500 rpm for 2 min), washed with EtOH for three times, and then dried in an oven under vacuum at 60 °C overnight.

### Characterization

PXRD patterns were recorded using a BRUKER D2 PHASER diffractometer equipped with a Cu sealed tube (λ = 1.54184 Å). The diffraction patterns were scanned at 30 kV and 10 mA at ambient temperature with a scan speed of 0.1 sec/step, a step size of 0.02° in 2θ, and a 2θ range of 5°–40°. Optical microscope images were collected by Nikon ECLPSE LV100NPOL. Fourier transform infrared spectra were collected using a BRUKER ALPHA spectrophotometer with a wavenumber region of 4000–400 $cm^{-1}$. Elemental analyses for C/H/O/N were measured in a Varian EL III element analyzer. Thermogravimetric analyses (TGA) were performed on an American TA TGA Q5000 differential thermal analyzer. The samples were heated in $N_2$ atmosphere (60 mL/min) from room temperature to 800 °C with a heating rate of 10 °C/min. Scanning electron microscopy and elemental mapping of EDS were carried out on a Hitachi S4800 field-emission scanning electron microscopy (FE-SEM) equipped with EDS. High-resolution transmission electron microscopy (TEM) were performed using a JEOL 2010 microscope operated at 200 kV. The emission spectra of TMOF-10-$NH_2$(I)/TMOF-10-$NH_2$(Br) samples were measured on a HORIBA FLUOROLOG-3 setup in reflection geometry. Time-resolved photoluminescence were performed at room temperature with a time-correlated single photon counting (TCSPC) technique on HORIBA FLUOROLOG-3. The excitation wavelength was 365 nm provided by an EPL-360PS pulsed diode laser. The lifetimes were calculated by fitting data to an exponential decay function using fluorescence decay analysis software. Temperature-dependent emission of TMOF-10-$NH_2$(I) was carried out on Edinburgh FLS1000 equipped with autotuning temperature controller. During the test, the temperature of sample chamber was gradually changed from 107 K to 297 K. The SPV spectra were performed on the basis of a lock-in amplifier (Sr830-D SP). The measurement systems include a computer, a light chopper (SR540), monochromatic light, and a sample cell. The monochromatic light was generated by a 500 W Xenon lamp (CHFXQ500 W, Global Xenon Lamp Power) with a grating monochromator (Omni-3007, no. 16047, Zolix). Femtosecond TA spectroscopy were obtained on a Helios pump-probe system (Ultrafast Systems LLC) combined with an amplified femtosecond laser system (Coherent). Metallic elemental contents for Ru and Pb were performed on a Perkin Elmer Optima 8300 ICP-OES.

### Single crystal X-ray diffraction

A suitable single crystal of TMOF-10-$NH_2$(I) and TMOF-10-$NH_2$(Br) was selected under an optical microscope (NIKON ECLIPSE LV100N POL) and mounted onto a glass fiber. The diffraction data of the pristine single crystal were collected at ambient temperature using graphite-monochromated Mo-Kα radiation (λ = 0.71073 Å), which was operated at 50 kV and 30 mA on a Bruker SMART APEX II CCD area detector X-ray diffractometer. The diffraction scan method includes a combination of phi and omega scans with the scan speeds of 3 s/° for the phi scans and 1 s/° for the omega scans. The crystal structure was solved by direct methods and expanded routinely. The model was refined by full-matrix least-squares analysis of $F^2$ against all reflections. All non-hydrogen atoms were refined with anisotropic thermal displacement parameters. Softwares for crystal structure analysis include APEX3 v2018.1, SHELXTL v6.14, and DIAMOND v4.6.1. The related details of crystallographic data and structural refinement are summerized in Supplementary Tables 1, 2. The simulated powder patterns were calculated by Mercury software using the crystallographic information file from the single-crystal X-ray diffraction data.

### UV-Vis diffusion reflectance spectroscopy

UV-Vis diffusion reflectance spectra in the range of 200–800 nm wavelength were collected at room temperature by a SHIMADZU UV−2600 spectrometer equipped with an integrating sphere. All test samples were grinded at μm-sized powders, then packed into the sample cell and used 100% $BaSO_4$ as a reference standard. Reflectance spectra were converted to absorption according to the equation:

$$A = 2 - lg(T\%) \tag{9}$$

A and T represent the absorbance and reflectance, respectively. The bandgaps of TMOF-10-$NH_2$ were obtained by extrapolation of the linear region of Tauc-plot.

## Chemical and thermal stability

Chemical stability experiments were carried out by incubating ~50 mg of TMOF-10·NH$_2$(I) and TMOF-10·NH$_2$(Br) as-synthesized crystals at different humidity conditions (50–90%) for 24 h. The samples were then collected for PXRD analysis and mass balance measurements. Thermal stability experiments were performed by treating as-synthesized TMOF-10·NH$_2$(I) and TMOF-10·NH$_2$(Br) in a static oven at 120 °C, 150 °C and 200 °C for 12 h. After cooling to the room temperature, PXRD and mass balance measurements were performed.

## Gas sorption

TMOF-10·NH$_2$(I) and TMOF-10·NH$_2$(Br) (*ca.* 50 mg) were activated by incubation in EtOH solvent at elevated temperature, followed by vacuum drying at 80 °C for overnight. The samples were then transferred in a pre-weighed analysis tube, heated at 80 °C under the outgas rate to <5 mm Hg for 600 min to remove all residual solvents on the Micromeritics ASAP 2020 absorption analyser. The sample tube was re-weighed to obtain a consistent dry mass for the degassed sample. Gas sorption isotherms were recorded volumetrically at 273 K and 298 K for CO$_2$, respectively. Isosteric heat of absorption (Qst) for CO$_2$ was calculated by applying the Clausius-Clapeyron equation:

$$\left(\frac{\delta \ln P}{\delta T}\right)_\theta = \frac{Qst}{RT^2} \tag{10}$$

in which *P* = equilibrium pressure, *T* = temperature, θ = absorption capacity divided by surface area, and *R* = 0.00831 kJ/mol·K.

## AC Hall measurement

Hall effect measurements were performed on the Accent HL5500 Hall System, in which samples were deposited on the glass substrate with four-point gold electrode contacts on each corner according to van der Pauw technique. The magnetic field for the test is 0.5 T and the range of test current was from −200 to 200 mA. The Hall system uses a rotating parallel dipole line magnet that generates AC field with pure harmonic, unidirectional, and strong magnetic field followed by Fourier spectral analysis and lock-in detection of the Hall signal.

## Photocatalytic CO$_2$ reduction

For photocatalytic CO$_2$ reduction with H$_2$O vapor, 10 mg of photocatalysts were uniformly dispersed on a quartz filter membrane (diameter 5 cm), which was then placed at the center of the homemade-sample holder. The holder was elevated from the bottom of the reaction cell to avoid the direct contact between water and photocatalyst. 5 mL degassed pure water were added to the reaction cell. The reaction system was evacuated to vacuum and refilled with high-purity CO$_2$ (99.99%) for three times to remove oxygen and other undesired gases, which was then pumped by atmospheric pressure of CO$_2$ that was bubbled through deionized water. The temperature of the reaction cell was controlled at 10 °C by recirculating cooling water system during irradiation. The light source for the photocatalysis was a 300 W Xe lamp (PLS-SXE300/300UV, Beijing Perfect light Technology Co., Ltd.). Upon the light irradiation, the product gases were quantitatively analyzed by online gas chromatograph (GC7860Plus, Shanghai nuoxi Instrument Co., Ltd.), with a flame ionization detector (FID) for determination of CO and CH$_4$ by identifying the chromatographic peaks. The consecutive three-cycle photocatalytic studies were performed by evacuation and refilling CO$_2$ for every 4 h.

To determine the water oxidation product, the volume of H$_2$O was increased to 30 mL, while the other conditions were kept to be the same as before. Every 4 h, 1 mL of the liquid was sampled for quantitative measurement, and the system was refilled with CO$_2$. To confirm the production and accumulation of H$_2$O$_2$, the concentration of H$_2$O$_2$ was quantified by colorimetry and DPD/POD method, respectively[60].

1. For the colorimetry method, the amount of H$_2$O$_2$ in the liquid phase was measured by UV-Vis spectra according to the color changes from Fe$^{2+}$ to Fe$^{3+}$ at wavelength of 330 nm as shown in the following equation:

$$Fe^{2+} + H_2O_2 + H^+ \rightarrow Fe^{3+} + 2H_2O \tag{11}$$

   Specifically, 0.1 mL of 0.1 M FeCl$_2$ in 1.0 M HCl aqueous solution was added to a mixed solution containing 1 mL of liquid sample and 0.9 mL of 1.0 M HCl aqueous solution. The H$_2$O$_2$ concentration was then measured by Fe$^{3+}$ colorimetry (330 nm).

2. For the DPD/POD method, 1 mL liquid sample was diluted to 2.5 mL with deionized water, which was then mixed with 400 μL 1 M KH$_2$PO$_4$/K$_3$PO$_4$ buffer solution. 50 μL *N,N*-diethyl-1,4-phenylene-diamine (DPD) and 50 μL horseridish peroxidase (POD) solution were then added. After coloration for 60 min, the mixture was analyzed by UV-Vis spectroscopy. A mixture of 2.5 mL deionized water with 400 μL buffer solution and 50 μL DPD was measured as the baseline. The stock solution of DPD was prepared by dissolving 0.1 g DPD in a 10 mL 0.05 M H$_2$SO$_4$ solution. The solution of POD was prepared by dissolving 10 mg POD in 10 mL deionized water and was kept in refrigerator for use.

## In situ DRIFTS

In situ DRIFTS over TMOF-10·NH$_2$(I) was performed on a Thermo Scientific Nicolet 6700FT spectrometer. The CO$_2$ flow was bubbled into deionized water and then passed through TMOF-10·NH$_2$(I) powders which loaded on the center of sample cell. The sample was irradiated by the Xenon lamp through a quartz window. The gaseous mixture of CO$_2$ and H$_2$O vapor were steadily produced before the measurement. After then, the IR spectra were collected in situ through the MCT detector.

## AQY test for photocatalytic CO$_2$ reduction

The AQY of CO$_2$-to-CO conversion was measured under specific excitation wavelength with a band pass filter (λ = 400, 425, 450, 475, 500, 525 nm), which was irradiated by the 300 W Xe lamp on the same setup. The lamp was fixed at a distance of 8 cm from the photocatalyst membrane. The number of the incident photons was measured by a radiant power energy meter (PL-MW2000 Photoradiometer, Perfect Light Co., Ltd.). The irradiated area was measured to be 0.000625 m$^2$ (2.5 cm × 2.5 cm). The temperature of the cell was fixed at 10 °C by a cooling water system. In general, the AQY of CO$_2$-to-CO conversion is calculated as follows:

$$AQY(\%) = \frac{\text{numbers of the elections taking part in reduction}}{\text{number of the incident photons}}$$
$$= \frac{N_{CO} \times 2 \times N_A}{P \times S \times \frac{\lambda}{hc}} \times 100\% \tag{12}$$

where $N_{CO}$ is the amount of CO, $N_A$ is the Avogadro's number, $P$ is the energy density of of the used AM1.5 G, $S$ is the irradiation area, $h$ is the Planck's constant, $c$ is the speed of light, $\lambda$ is the wavelength of the incident light.

## Electrochemical measurements

Photoelectrochemical studies were performed in a CHI 760E electrochemical work station (Shanghai Chenhua) in a standard three-electrode system with the photocatalyst-coated indium-tin oxide (ITO) as the working electrode, platinum plate as the counter electrode, and an Ag/Ag$^+$ electrode (photoresponsive signals as the reference electrode. A 0.1 M tetrabutylammonium hexafluorophosphate (TBAPF$_6$) in dichlormethane solution was used as the electrolyte. The as-synthesized samples (10 mg) were dispersed into a solution containing EtOH (1 mL) and Nafion (20 μL). The working electrodes were prepared by dropping the above suspension (100 μL) onto the surface of an ITO plate (1.0 cm × 1.0 cm) and dried at room temperature. Electrochemical impedance spectroscopy (EIS) was performed in the frequency range from 0.1 Hz to 1 MHz. A 300 W Xenon lamp was used

as the incident light in the photocurrent measurements, and photo-responsive signals were measured under transient light at 0.3 V.

## Computational methods

All the first-principles spin-polarized calculations were performed by using the Vienna ab initio Simulation Program (VASP)[78,79]. The generalized gradient approximation (GGA) in the Perdew-Burke-Ernzerhof (PBE) form and a cutoff energy of 500 eV for planewave basis set were adopted[80]. A $2 \times 2 \times 1$ Monkhorst-Pack[81] k grid was used for sampling the Brillouin zones at structure calculation, whereas a dense mesh of $5 \times 5 \times 1$ was used for the electronic structure calculations. The ion-electron interactions were described by the projector augmented wave (PAW) method[82]. A vacuum space exceeds 20 Å was employed to avoid the interaction between two periodic units. The convergence criteria of structure optimization were chosen as the maximum force on each atom less than 0.02 eV/Å with an energy change less than $1 \times 10^{-5}$ eV. The Grimme's DFT-D3 scheme of dispersion correction was adopted to describe the van der Waals (vdW) interactions in these systems[83].

The standard hydrogen electrode model proposed by Nørskov and co-workers was employed to calculate the Gibbs free energy change ($\Delta G$) for each elemental step[84,85]. The $\Delta G$ is defined as shown in equation:

$$\triangle G = \triangle E + \triangle ZPE - T \triangle S + \triangle G_U + \triangle G_{pH} \qquad (13)$$

in which $\Delta E$ is the adsorption energy based on DFT calculations, $\Delta ZPE$ is the zero-point energy correction, $T$ is the temperature, $\Delta S$ is the entropy change, $U$ stands for the applied electrode potential, and $\Delta G_{pH}$ represents the free energy correction of the pH (pH = 0 is assumed in the acidic medium in this study).

## Data availability

The data supporting the findings of this study are available in the paper and Supplementary Information files. The X-ray crystallographic coordinates for structures reported in this study have been deposited at the Cambridge Crystallographic Data Centre (CCDC), under deposition numbers 2124593 (TMOF-10-NH₂(Br)), 2124591 (TMOF-10-NH₂(I)) and 2175446 ([Pb(NH₂-bdc)]ₙ). These data can be obtained free of charge from The Cambridge Crystallographic Data Centre via www.ccdc.cam.ac.uk/data_request/cif. Source data are provided with this paper.

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

## Acknowledgements
This work was supported by grants from the National Natural Science Foundation of China (21971197, 22171214 and 22076144), the Shanghai Rising-Star Program (20QA1409500), the Natural Science Foundation of Shanghai (22ZR1463200), the Recruitment of Global Youth Experts by China and the Science & Technology Commission of Shanghai Municipality (19DZ2271500 and 20230712200).

## Author contributions
X.C. and H.F. conceived the project. H.F. supervised the work and X.C. performed the vast majority of experimental studies. C.P. contributed in the photocatalyst synthesis, W.D. assisted in the crystallographic analysis, and L.Y. assisted in synthesis and photophysical studies. X.C., Y.W. and H.F. wrote and revised the paper. All authors contributed to the manuscript preparation.

## Competing interests
The authors declare no competing interests.
