## [Peer Review File · Nature Communications]

Title: Bromo- and Iodo-Bridged Building Units in Metal-Organic Frameworks for Enhanced Carrier Transport and CO₂ Photoreduction by Water VaporREVIEWER COMMENTS

Reviewer #1 (Remarks to the Author):

By the method of crystal engineering, the authors realized the combination of MOFs and PbX_2 ($X = \text{Br}, \text{I}$), namely, TMOF-10-NH₂(Br) and TMOF-10-NH₂(I). The as-synthesized materials feature the open frameworks, which consist of 1D $[\text{Pb}_2\text{Br}]^{3+}$ or $[\text{Pb}_2\text{I}]^{3+}$ chains as SBUs and 2-aminoterephthalic acid as linkers. In contrast to lead perovskites, the $[\text{Pb}_2\text{X}]^{3+}$ -based MOFs in this paper have accessible, well-defined porosity and high moisture stability for gas-phase photocatalytic reaction between CO₂ and H₂O. Although this work looks solid, there are still many problems that need to be solved. Therefore, I would like not to recommend its publication in Nature Communication. Instead, some other journals, such as ACS Catalysis, Chemical Science or Journal of Materials Chemistry A, may be suitable.

Some detailed comments are listed below:

- 1) Many factors (e.g., structure, band gap, light absorption coefficient, electron/hole mobility and stability) may have a large effect on the CO₂ photoreduction of as-synthesized materials. Structurally, although the authors conduct a comparative study with UiO-66-NH₂ as an example, this is not very meaningful due to the large structural discrepancy. From the DFT results, we can see that the halogen make insignificant contribution to the band gap and frontier orbital. Therefore, a comparative study using the similar Pb-based instead of PbX-based MOFs materials seem to be convincing. Most mentioned-above factors should be discussed. To the best of our knowledge, their synergy may be the main reason why they exhibit good carrier transport and excellent CO₂ photoreduction. More detailed studies may be required.
- 2) Recently, it has been found that effective mass may also have a large effect on the carrier transport. Is this consistent with the existing conclusions? Some related discussion had better be provided.
- 3) In the preparation of title compounds, the perchloric acid was used, which has often appeared in some previous work, such as *Angew. Chem. Int. Ed.* 2017, 56, 14411–14416; *Chem. Sci.* 2018, 9, 1627–1633; *J. Am. Chem. Soc.* 2012, 134, 10729–10732 and *Cryst. Growth & Des.* 2010, 10, 823–829. What is the role?
- 4) There have been some CO₂ reaction studies using stable MOFs and perovskite materials (*ChemSusChem* 2019, 12, 4769–4774; *J. Mater. Chem A* 2019, 7, 13762–13769; *Appl. Catal. B* 2017, 210, 131–140; *ACS Energy Lett.* 2018, 3, 2656–2662; *J. Phys. Chem. Lett.* 2019, 10, 7965–7969), what are the advantages of the title compounds? What level? How is the efficiency and capability? The authors should comment it comprehensively in the revised manuscript.
- 5) In Figure 2a, there are two unnecessary diffraction peaks at about 32° and 38°, respectively. Why?
- 6) Some references about the sentences “For example, the UV-absorbing Zr₆O₄(OH)₄ clusters are the active sites for CO₂ photoreduction in the benchmark UiO-66~68 MOFs.” and “In addition, the conventional metaloxo SBUs in MOFs have very confined carrier mobility and carrier diffusion lengths, analogous to typical metal oxide semiconductors.” are missing.
- 7) To better understand the CO₂ photoreduction behavior of title compounds, the photopotential studies, flat-band potential/conduction band energy evaluation and carrier density seem to be necessary. Some supplementary experiments and discussion should be added.
- 8) Before dealing with UV-Vis absorption spectra, it is necessary to first clarify whether the compound is

a semiconductor with a direct band gap or an indirect band gap?

9) The quality of the DFT band structure is poor. The bands are flat and quite meaningless due to the quality of the calculation. This part needs to be improved.

10) There are also some grammar, spelling and format mistakes in the manuscript, such as:

(a) In page 2, line 6, "X= Br-/I-" should be "X = Br-/I-";

(b) In page 5, line 11, "visble" should be "visible";

(c) In page 8, line 3, "analysis" should be "analyses";

(d) ref 9, 21, 42, 54, 57, "Angew. Chem." should be "Angew. Chem. Int. Ed.";

Reviewer #2 (Remarks to the Author):

There is a natural connection between photovoltaic and photocatalyst, because both involves the utilization of solar energy to power electronics or drive chemical reactions. One of the-state-of-the-art solar cell systems is materials with perovskite structure, where lead halide is the inorganic component. It is ideal that if this structure can be used for photocatalytic conversion of CO₂, due to the excellent light absorption efficiency and wavelength coverage. However, this perovskite is usually moisture sensitive, making it hard to be applied for photocatalysis, where the presence of water is often inevitable. This work avoided the direct use of perovskite in CO₂ photoreduction, but extracting the core lead halide unit, as the building block to construct MOFs, an alternative and smart way with the attempt to circumvent the moisture sensitive issue. The resulting MOF is composed of linear lead halide SBUs, which was rare for MOFs. The covalently bonded MOFs offers higher stability and porosity than the ionic bonded halide perovskites. At the same time, both the light absorption and electron conduction feature of perovskite is largely maintained, reflected in good charge carrier mobility and relatively long lifetime of the excited electrons; better than those of conventional metal oxide based MOFs. These lead halid MOFs exhibit high photocatalytic activity for gas-phase CO₂ reduction in the presence of water vapor. This is an excellent example that the light adsorption site is incorporated into MOF structure with the promotion in charge separation. Although molecular oxygen is yet generated, this method stands as a new direction to tailor MOFs with better photocatalytic properties, showing potential for future research. In general, the synthesis is novel, the structure characterization is solid, and extensive spectroscopic analysis gives insight into the mechanism, worth of highly recommendation for publication in Nat. Commun, after the following suggestions are fully considered.

1. The success in gas phase CO₂ photoreduction was quite rare, especially in the presence of water vapor, with only very few report with excellent charge separation, eg. Nature, 2020, 586, 549. This is one of the highlight in the current work that worth of further discussion. In most of the previous works, CO₂ reduction is carried out in solution, where the cycling and production collection become issues.

2. The halide species in perovskite structure play critical role. Is it the same for TMOFs here? What is the reason behind the better performance show in I version than the Br version?

3. The balance between the reduction side and oxidation side in the overall CO₂ reduction is critical. One of a key issue for the use of MOFs for CO₂ reduction is the accumulation of holes, once the electrons are consumed faster, therefore will lead to gradual self-oxidation. Although the oxidized and

reduced species generated is not yet important at this initial stage, the quantitative amount will provide critical information about the charge separation and utilization. Is the consumption of electrons and holes are equivalent in terms of products? What is the cycling performance?

Minor suggestion:

4. The crystal surface might also influence the catalytic performance. In the page 9, the author mentioned “the (001) facets, which is the major exposed crystal facet of TMOF-10-NH₂(I)”. What is the detailed method to determine the exposed crystal surface? How does it influence the performance? Control experiments might be necessary.

Control experiments might be necessary.

5. IR studies are informative to the chemical structure. It is recommended the performance of FT-IR spectra for MOFs after photocatalysis to investigate the leaching/change of organic linkers.

6. For the index of PXRD pattern, hkl values are directly used without parenthesis. This is a common mistake that can even be found in some of the textbooks.

7. Typos in the manuscript should be fixed, eg, in the Fig. 6b, the “CO₂” was incorrectly labelled as “CO²”.

8. Surface areas of these MOF should also be discussed.

9. There are rigorous standards in the test of AQE. At least, the distance between the lamp and sample, size of the irradiated area, and temperature of the cell should be fixed before an accurate value is estimated. The AQE value reported here is not the pure MOF, but with Ru as dopant, this should be specified in the text, otherwise, it is confusing.

Reviewer #3 (Remarks to the Author):

In this work, the authors report the preparation and photocatalytic application of a Metal-Organic framework comprised of Lead-Bromo/Iodo Secondary Building Units and 2-amino-teraphtalate. The authors consistently characterize the hybrid materials and thoroughly investigate their photocatalytic activity for CO₂ reduction, giving emphasis on the mechanistic aspects. This work is of great significance, showcasing how combining the chemistry of two distinct class of materials (MOFs and Organo-Lead Halide materials) can give rise to hybrid materials with enhanced properties. Such strategy can further diversify the portfolio of efficient photocatalysts and advance this field. I find this detailed research study very interesting, however there are few inconsistencies that need to be addressed. I thus recommend this manuscript for publication in Nature Communications after the following minor issues are addressed:

1) There are several spelling/grammatical mistakes in the manuscript (especially in the methods section) that need to be corrected. I would thus recommend proofreading.

2) There are references missing at the end of the second paragraph on page 5: ‘Their photocatalytic acitivity in CO₂ reduction (AQE of 1.4% at 400 nm) is the highest reported value in organolead halide hybrids when using water vapor as the sacrificial agent, and higher than many benchmark metal-oxo MOFs with the identical organic linker (e.g. UiO-66(Zr)-NH₂ and MIL-101(Fe)-NH₂) as well.’

3) On page 4 the authors mention: 'For example, the UV-absorbing $Zr_6O_4(OH)_4$ clusters are the active sites for CO_2 photoreduction in the benchmark UiO-66~68 MOFs. Employing the light-absorbing linkers and/or guest molecules will introduce new charge transfer problems, such as the energetically unfavorable ligand-to-node charge transfer and noncovalent interface charge transfer, respectively.'. In the UiO-66 and UiO-68 series, the light absorption is governed by that of the ligand, not of the SBUs. In addition, introduction of light-harvesting linkers is not problematic, but a promising strategy for enhancing the optoelectronic properties of MOF photocatalysts. I would recommend rephrasing this section, as it is very confusing.

4) The caption of Figure 2, should be modified, as it does not illustrate the stability, but the PXRD patterns of the as-synthesized materials.

5) There are inconsistencies with the terminology used. More specifically, 'RE' should be replaced by 'RH' on page 2, the term 'AQE' should be replaced by 'AQY' on page 34, 'Volts' should be replaced by 'eV' on page 12, and 'protonation' should be replaced by 'deprotonation' on page 21. In addition, the authors mention on page 8 that there are '(CH₃)₂NH₃⁺' guests, referring to Figure S13. Do the authors mean dimethylammonium or perhaps dimethylamine? If there are indeed cationic species, this should influence the chemical formula of the MOFs. This should be clarified and properly discussed.

6) The authors state 'The high-energy emission is ascribed to be the charge transfer from bdc-NH₂ to Pb²⁺ centers, which will be discussed later.'. Please provide references of research studies in the literature or experiments/calculations supporting this argument.

7) The timescales for TAS and TPL are very different; ps and ns for TAS and TPL, respectively, which seems counterintuitive, as through TAS all the charge carriers can be tracked down, including the ones radiatively recombining with the holes (PL). How do the authors explain that? I would recommend a more detailed description of the insights from the TAS and TPL experiments to be added in the manuscript.

8) The authors should provide the BET surface areas of the MOFs, as the high porosity of the reported materials is emphasized.

Point-by-point response letter

We thank all the reviewers for their insightful comments and suggestions, which contribute to the enhanced quality of our manuscript. We have thoroughly revised the manuscript with additional experimental results and discussion according to reviewers' comments, which are highlighted in the manuscript.

Reviewer 1

By the method of crystal engineering, the authors realized the combination of MOFs and PbX_2 ($X = Br, I$), namely, TMOF-10-NH₂(Br) and TMOF-10-NH₂(I). The as-synthesized materials feature the open frameworks, which consist of 1D $[Pb_2Br]^{3+}$ or $[Pb_2I]^{3+}$ chains as SBUs and 2-aminoterephthalic acid as linkers. In contrast to lead perovskites, the $[Pb_2X]^{3+}$ -based MOFs in this paper have accessible, well-defined porosity and high moisture stability for gas-phase photocatalytic reaction between CO₂ and H₂O. Although this work looks solid, there are still many problems that need to be solved. Therefore, I would like not to recommend its publication in Nature Communication. Instead, some other journals, such as ACS Catalysis, Chemical Science or Journal of Materials Chemistry A, may be suitable.

- We thank the reviewer for the comments. The control experiments using Pb-based MOF and the advantages of the title compounds over stable MOFs/perovskites have been clarified. In addition, the quality of DFT calculations has been substantially improved, and effective mass studies have been added to the manuscript. We hope our additional experiments and explanations (described below) will resolve any confusion.
- This is the first study to use crystal engineering to combine the advantages of two distinctive class of crystalline materials (*i.e.* MOFs and organolead halide) for overall CO₂ photoreduction and H₂O oxidation. The vast majority of MOFs consist of metal-oxo SBUs, therefore this first generation of metal-bromo/iodo MOFs are amenable for synthetic design and pave the way for constructing a potentially general class of photocatalytic materials.

Some detailed comments are listed below:

1. Many factors (e.g., structure, band gap, light absorption coefficient, electron/hole mobility and stability) may have a large effect on the CO₂ photoreduction of as-synthesized materials. Structurally, although the authors conduct a comparative study with UiO-66-NH₂ as an example, this is not very meaningful due to the large structural discrepancy. From the DFT results, we can see that the halogen make insignificant contribution to the band gap and frontier orbital. Therefore, a comparative study using the similar Pb-based instead of PbX-based MOFs materials seem to be convincing. Most mentioned-above factors should be discussed. To the best of our knowledge, their synergy may be the main reason why they exhibit good carrier transport and excellent CO₂ photoreduction. More detailed studies may be required.

- We thank the reviewer for the important suggestion, and agree with the reviewer that the direct comparison with Pb-based MOF is necessary. Light absorption and carrier transport are two important photophysical properties for efficient photocatalysis. In this study, the halide species do not make a significant contribution to tune the bandgaps for light absorption, but are of key importance to enhance the intrinsic carrier transport properties of TMOFs. The metal-bromo/iodo SBUs in TMOFs afford the enhanced carrier transport properties, superior to conventional metal-oxo SBUs. Indeed, the

comparative study between Pb-based MOF and PbX-based MOF again confirms this conclusion. We hope our additional experiments and explanations (described below) will resolve any confusion.

- First, we have successfully synthesized a Pb^{2+} -based MOF with the chemical formula of $[\text{Pb}(\text{NH}_2\text{-bdc})]_n$, using the identical organic linker as TMOF-10-NH₂ but not involving halide species. X-ray crystallography of $[\text{Pb}(\text{NH}_2\text{-bdc})]_n$ reveals that Pb^{2+} centers and carboxylates in NH₂-bdc are connected into a 1D chain, which are six-connected by organic struts into a porous network (**Figure S30**). Both TMOF-10-NH₂ and $[\text{Pb}(\text{NH}_2\text{-bdc})]_n$ occupy 1D linear SBUs and 1D pore channels, showing analogous behaviors in structural topology.
- UV-Vis absorption spectroscopy and Tauc plot show a bandgap of 2.43 eV for $[\text{Pb}(\text{NH}_2\text{-bdc})]_n$, which is close to the bandgaps of TMOF-10-NH₂(Br/I) (**Figure S32**). However, the absorption tail of $[\text{Pb}(\text{NH}_2\text{-bdc})]_n$ is lower than TMOF-10-NH₂, again confirming the tail absorption of TMOFs originating from the electron-phonon coupling of 1D metal halide SBUs.
- Despite the analogous bandgaps between Pb-based MOF and PbX-based MOF, the carrier transport characteristics of TMOF-10-NH₂ are superior to that of $[\text{Pb}(\text{NH}_2\text{-bdc})]_n$. Surface photovoltage spectroscopy (SPV) indicates a very weak photovoltage response at ~370 nm for $[\text{Pb}(\text{NH}_2\text{-bdc})]_n$, indicate a less favorable carrier transport (**Figure S33**). In addition, the steady-state photoluminescence study of $[\text{Pb}(\text{NH}_2\text{-bdc})]_n$ indicates an emission centered at 455 nm (**Figure S34**). Time-resolved photoluminescence decay shows an average decay lifetime of 0.35 ns (**Figure S35**), substantially lower than the average lifetime of TMOF-10-NH₂(Br/I) (11.92~15.31 ns) (**Figure S28**). These photophysical studies verify the the halide species are critical to improve the carrier transfer property of MOFs.
- The control experiments of photocatalytic CO₂ reduction have also been performed under the identical reaction conditions (**Figure 5f**). The CO evolution rate is determined to be 5.2 $\mu\text{mol h}^{-1} \text{g}^{-1}$ for $[\text{Pb}(\text{NH}_2\text{-bdc})]_n$, which is not only substantially lower than TMOF-10-NH₂ but also lower than MIL-101(Fe)-NH₂ and UiO-66(Zr)-NH₂. To note, the crystallinity of $[\text{Pb}(\text{NH}_2\text{-bdc})]_n$ is well retained throughout the photocatalysis (**Figure S49**), excluding the possibility of decreased performance from stability issue. Therefore, it is probably ascribed to lower photocatalytic activity of Pb^{2+} -oxo sites than the benchmark Zr^{4+} -oxo/ Fe^{3+} -oxo clusters in MOFs, despite occupying the analogous carrier transport properties. These photocatalytic studies again suggest the importance of the bridging halide species in linear SBUs to realize the intrinsic carrier transport characteristics in MOFs.
- Besides the control experiments of $[\text{Pb}(\text{NH}_2\text{-bdc})]_n$, our comparative studies between TMOF-10-NH₂(Br) and TMOF-10-NH₂(I) rationalize the significant role of bridging halide species of SBUs in photocarrier transport. Both materials have analogous band structures, but the carrier transport of TMOF-10-NH₂(I) is more favorable than TMOF-10-NH₂(Br) based on SPV, TAS, Hall effect and TPL studies. The trend is consistent with photocatalytic CO₂ reduction performances, which is ascribed to the intrinsic soft nature of bridging iodide that has been studied in perovskites.

2. Recently, it has been found that effective mass may also have a large effect on the carrier transport. Is this consistent with the existing conclusions? Some related discussion had better be provided.

- We thank the reviewer for this valuable suggestion. It is widely accepted that smaller carrier effective masses afford higher carrier mobilities and longer carrier diffusion lengths. On the basis of the parabolic approximation, the effective mass (m^*) of carriers existing around the bottom of the conduction band or the top of the valence band are estimated as the fitting of the dispersion relation:

$$m^* = \hbar^2 \left[\frac{\partial^2 \varepsilon(k)}{\partial k^2} \right]^{-1}$$

where $\varepsilon(k)$ are the band edge eigenvalues and k is the wavevector. Based on above equation and the band structure of TMOF-10-NH₂(I), the effective masses (m_e^* and m_h^*) of TMOF-10-NH₂(I) are calculated to be $0.67m_0$ and $0.18m_0$, respectively. Both values are substantially lower than the vast majority of benchmark metal-oxo MOFs (e.g. m_e^* : $4.2m_0$ and m_h^* : $84.2m_0$ for MIL-125(Ti)-NH₂; m_e^* : $67.4m_0$ and m_h^* : $117.8m_0$ for MOF-5(Zn)-NH₂ and m_e^* : $0.8m_0$ and m_h^* : $104.9m_0$ for UiO-66(Zr)-NH₂) (*Adv. Funct. Mater.* **2020**, 30, 2003792).

- Organolead halide hybrids generally possess low effective mass due to the strong spin-orbit coupling of Pb $6p$ orbitals and dispersive VBM/CBM (*Adv. Mater.* **2019**, 31, 1803792; *Nat. Rev. Mater.* **2016**, 1, 15007). Our study indicates that the effective masses of TMOF-10-NH₂(I) are comparable to organolead halide perovskites (m_e^* : $0.17\sim 0.73m_0$ and m_h^* : $0.28\sim 0.36m_0$ for MAPbI₃), suggesting our porous halide-bridged MOFs maintain the excellent carrier transport characteristics of lead halide hybrids.

3. In the preparation of title compounds, the perchloric acid was used, which has often appeared in some previous work, such as *Angew. Chem. Int. Ed.* 2017, 56, 14411–14416; *Chem. Sci.* 2018, 9, 1627–1633; *J. Am. Chem. Soc.* 2012, 134, 10729–10732 and *Cryst. Growth & Des.* 2010, 10, 823–829. What is the role?

- We thank the reviewer for the comment. First, the perchloric acid performs as an inorganic acid to regulate the pH of the synthetic condition. In general, a strong acidic environment is essential for crystal growth of our title compounds, affording the organocarboxylic acid in the protonated form during synthesis. Our attempts to perform the synthesis in neutral or basic conditions have not been successful to achieve the title MOFs. Second, the counter anion is also important to choose the acid. For example, hydrochloric/hydrobromic acid will introduce a second halide into the final MOF products. The counter anions of many other inorganic acids, such as HNO₃ and H₂SO₄, have stronger coordination ability than the perchloric acid, which may interfere the crystallization process between PbX species and organocarboxylic acid. Finally, the perchloric acid has been known as a crystallization stabilizer, analogous to the hydrofluoric acid acting as a mineralizer in zeolite synthesis (*J. Phys. Chem. B* **1998**, 102, 4147). Both of them are not present in the final synthesized material, but help the phase formation of crystallization.

4. There have been some CO₂ reaction studies using stable MOFs and perovskite materials (*ChemSusChem* 2019, 12, 4769–4774; *J. Mater. Chem A* 2019, 7, 13762–13769; *Appl. Catal. B* 2017, 210, 131–140; *ACS Energy Lett.* 2018, 3, 2656–2662; *J. Phys. Chem. Lett.* 2019, 10, 7965–7969), what are the advantages of the title compounds? What level? How is the efficiency and capability? The authors should comment it comprehensively in the revised manuscript.

- We agree with the reviewer that stable MOFs, stable perovskites and stable MOF/perovskite composites have been studied in photocatalytic CO₂ reduction. Herein, we discuss the advantages of our TMOFs over these three classes of stable

photocatalysts, and hope our explanations (described below) will resolve any confusion.

- **Comparison with stable MOFs.** The low carrier-transport characteristics remain to be a great challenge for conventional metal-oxo MOF photocatalysts based on typical carboxylate linkers (*Chem. Soc. Rev.* **2012**, 41, 115; *Chem. Soc. Rev.* **2017**, 46, 3185). A variety of synthetic approaches have been developed to use non-carboxylate linkers, often involving complicated synthesis. Herein, the carrier transport characteristics of our metal-bromo/iodo MOFs have been substantially improved, using a typical and simple NH₂-bdc strut. This leads to the enhanced photocatalytic activity of our TMOFs over many benchmark MOFs based on the identical NH₂-bdc linkers (**Figure 5f**). To note, this first generation of halide-bridged MOFs are amenable to synthetic design, allowing for the introduction of a variety of photoactive linkers and further improving the photocatalytic properties.
- **Comparison with stable perovskites.** The vast majority of organolead halide perovskites are susceptible to degradation upon the treatment of high-polarity protic molecules, owing to their ionic structures and the hydrophilic nature of organoammonium cations. There are only a few purely inorganic metal halide perovskites (*ChemSusChem* **2019**, 12, 4769) that are stable in moisture and water, but they occupy lower degree of structural tunability due to the absence of organic linkers. Our MOFs are coordination polymers and have higher structural stability than ionic structures as for perovskites. Moreover, lead halide MOFs is a more versatile platform to have organic linkers to tune functionality, which are not accessible in purely inorganic metal halide perovskites.
- **Comparison with MOF/perovskites composites.** The encapsulation of perovskite nanocrystals into a solid-state matrix (*e.g.* MOFs) is a straightforward approach to enhance stability and perform photocatalysis. However, this may raise some potential problems, which have been summarized by Julian A. Steele *et. al.* in a recent review (*ACS Energy Lett.* **2020**, 5, 1107). For example, the light may be blocked by the protective MOFs to reach the encapsulating perovskites. The new perovskite/MOF interface may interfere the charge transfer process. Realizing intrinsic stability for metal halide hybrids is an ideal scenario to perform long-term photocatalytic performance, which is the focus of our title compounds.
- Based on above-mentioned comparisons, our lead halide-based MOFs show four advantages, including moisture stability, porosity, strong light absorption and excellent carrier transport. The combination of all four characteristics has not been realized using an intrinsic single-component MOF or perovskite material. This affords our MOFs to be one of the highest reported performances for gas-phase CO₂ photoreduction in the presence of water vapor (**Table S3 and S4**), not to mentioned this first generation of halide-bridged MOFs are amenable to synthetic design for further improvements.

5. In Figure 2a, there are two unnecessary diffraction peaks at about 32° and 38°, respectively. Why?

- We thank the reviewer for comment. Indeed, both peaks at ~32° and ~38° correspond well to the simulated PXRD from single-crystal data (**Figure C1**). The differences in relative intensities between the experimental and simulated patterns are due to the nonrandom crystal orientations, which are commonly observed in MOF crystals. Sufficient manual grinding of MOF crystals into microcrystalline powders will avoid the issue, and the PXRD patterns have been revised accordingly in the manuscript (**Figure 2a**).

Figure C1. The experimental and simulated PXRD patterns for TMOF-10-NH₂ (right), and the partial enlarge figure for TMOF-10-NH₂(Br) in the range of 25°~40°.

6. Some references about the sentences “For example, the UV-absorbing Zr₆O₄(OH)₄ clusters are the active sites for CO₂ photoreduction in the benchmark UiO-66~68 MOFs.” and “In addition, the conventional metaloxo SBUs in MOFs have very confined carrier mobility and carrier diffusion lengths, analogous to typical metal oxide semiconductors.” are missing.

- We thank the reviewer for raising this point, and have added references and rephrased the sentences in the revised manuscript.
- The main texts and references are revised as follows: “For example, the UV-absorbing Zr₆O₄(OH)₄ clusters are the active sites for CO₂ photoreduction in the benchmark UiO-series MOFs (*Chem Eur. J.*, **2010**, 16, 11133). The introduction of light-harvesting organic linkers and/or guest molecules are necessary to achieve visible-light-driven photocatalysis (*Chem. Soc. Rev.*, **2014**, 43, 5982; *Acc. Chem. Res.*, **2019**, 52, 356). The conventional metal-oxo SBUs in MOFs have very confined carrier mobility and charge transport properties (*Chem. Soc. Rev.* **2012**, 41, 115; *Chem. Soc. Rev.* **2017**, 46, 3185).”

7. To better understand the CO₂ photoreduction behavior of title compounds, the photopotential studies, flat-band potential/conduction band energy evaluation and carrier density seem to be necessary. Some supplementary experiments and discussion should be added.

- The flat-band potentials are measured *via* Mott–Schottky plots for both TMOF-10-NH₂(Br) and TMOF-10-NH₂(I). The positive slopes of the plots again suggest the *n*-type semiconductive nature of TMOF-10-NH₂ for both materials, agreeing with the Hall effect measurement (**Figure S23**). The flat-band potentials are evaluated to be −1.11 V vs. Ag/AgCl electrodes (−0.91 V vs. NHE, pH 7) for TMOF-10-NH₂(I) and −1.37 V vs. Ag/AgCl electrodes (−1.17 V vs. NHE, pH 7) for TMOF-10-NH₂(Br). Both values are more negative than the redox potentials of CO/CO₂ (−0.48 V vs NHE, pH 7). It is generally known that the CBM of an *n*-type semiconductor is ~0.2 V more negative than the flat-band potential below the CBM (*Adv. Mater.* **2018**, 30, 1803401). Therefore, the CBMs of TMOF-10-NH₂(I) and TMOF-10-NH₂(Br) are estimated to be ~−1.11 V vs. NHE and ~−1.37 V vs. NHE, respectively. Both values are basically in line with the band positions measured by VB-XPS.
- Hall effect measurement indicates the carrier density/concentration to be 1.28×10¹⁵ cm^{−3} for TMOF-10-NH₂(I) and 8.87×10¹⁴ cm^{−3} for TMOF-10-NH₂(Br), respectively (Table 1). The higher carrier concentration of TMOF-10-NH₂(I) over TMOF-10-

NH₂(Br) again suggests that the iodide-bridged SBUs occupy more favorable carrier transport than the bromide counterpart.

- Owing to the pandemic lockdown in Shanghai, we are unable to access the instruments for photopotential studies. Nevertheless, we believe the photophysical properties of TMOF-10-NH₂ materials have been thoroughly clarified by a wide variety of experimental techniques (SPV, TA, Hall effect measurement, transient-state PL and flat-band potentials) as well as DFT calculations.

8. *Before dealing with UV-Vis absorption spectra, it is necessary to first clarify whether the compound is a semiconductor with a direct band gap or an indirect band gap?*

- We thank the reviewer for the kind suggestion. The direct or indirect bandgap of a semiconductor is verified by Tauc plots (*i.e.*, the curve of converted $(\alpha h\nu)^r$ versus $h\nu$ from the UV-Vis absorption spectra, in which α , h , and ν are the absorption coefficient, Planck constant, and light frequency, respectively). A direct bandgap material occupies the r value of 2, while an indirect bandgap material has the r value of 1/2. Both of our TMOF-10-NH₂ show good linear fits using $r = 2$, suggesting direct bandgaps for both TMOF-10-NH₂(Br) and TMOF-10-NH₂(I) (**Figure S18**).

9. *The quality of the DFT band structure is poor. The bands are flat and quite meaning less due to the quality of the calculation. This part needs to be improved.*

- We thank the reviewer for the comment. DFT calculations for the band structures have been performed again and the quality of the calculation has been improved (**Figure S25**). The relatively flat bands have been observed in many MOF structures (*J. Phys. Chem. C* **2014**, 118, 4567; *Adv. Sci.* **2021**, 8, 2100548; *Nano Lett.* **2018**, 18, 5596; *J. Am. Chem. Soc.* **2020**, 142, 12515), since their organic linkers (instead of inorganic units) make an important contribution to frontier orbitals.

10. *There are also some grammar, spelling and format mistakes in the manuscript, such as:*

(a) *In page 2, line 6, “X= Br-/I-” should be “X = Br-/I-”;*

(b) *In page 5, line 11, “visble” should be “visible”;*

(c) *In page 8, line 3, “analysis” should be “analyses”;*

(d) *ref 9, 21, 42, 54, 57, “Angew. Chem.” should be “Angew. Chem. Int. Ed.”;*

- We have carefully revised the manuscript to correct the grammar, spelling and format mistakes.

Reviewer 2

There is a natural connection between photovoltaic and photocatalyst, because both involves the utilization of solar energy to power electronics or drive chemical reactions. One of the state-of-the-art solar cell systems is materials with perovskite structure, where lead halide is the inorganic component. It is ideal that if this structure can be used for photocatalytic conversion of CO₂, due to the excellent light absorption efficiency and wavelength coverage. However, this perovskite is usually moisture sensitive, making it hard to be applied for photocatalysis, where the presence of water is often inevitable. This work avoided the direct use of perovskite in CO₂ photoreduction, but extracting the core lead halide unit, as the building block to construct MOFs, an alternative and smart way with the attempt to circumvent the moisture sensitive issue. The resulting MOF is composed of linear lead halide SBUs, which was rare for MOFs. The covalently bonded MOFs offers higher stability and porosity than the ionic bonded halide perovskites. At the same time, both the light absorption and electron

conduction feature of perovskite is largely maintained, reflected in good charge carrier mobility and relatively long lifetime of the excited electrons; better than those of conventional metal oxide based MOFs. These lead halid MOFs exhibit high photocatalytic activity for gas-phase CO₂ reduction in the presence of water vapor. This is an excellent example that the light adsorption site is incorporated into MOF structure with the promotion in charge separation. Although molecular oxygen is yet generated, this method stands as a new direction to tailor MOFs with better photocatalytic properties, showing potential for future research. In general, the synthesis is novel, the structure characterization is solid, and extensive spectroscopic analysis gives insight into the mechanism, worth of highly recommendation for publication in Nat. Commun, after the following suggestions are fully considered.

- We thank the reviewer for the strong endorsement.

1. The success in gas phase CO₂ photoreduction was quite rare, especially in the presence of water vapor, with only very few report with excellent charge separation, eg. Nature, 2020, 586, 549. This is one of the highlight in the current work that worth of further discussion. In most of the previous works, CO₂ reduction is carried out in solution, where the cycling and production collection become issues.

- We agree with the reviewer in terms of the importance in gas-phase CO₂ photoreduction and the recent TiO₂/MIL-101(Cr) study.
- The discussion and reference are revised as follows on page 3: “Deng and co-workers discovers the internal growth of TiO₂ inside a mesoporous MOF to generate molecular compartments, achieving efficient gas-phase CO₂ photoreduction in the presence of water vapor (*Nature*, **2020**, 586, 549).”

2. The halide species in perovskite structure play critical role. Is it the same for TMOFs here? What is the reason behind the better performance show in I version than the Br version?

- We thank the reviewer for the comment. The halide species in perovskite play an important role in both light harvesting and carrier transport. In our lead halide-based MOFs, the essential contribution of halide species to the intrinsic carrier transport is analogous to perovskites. The photophysical studies demonstrate high carrier mobility and long carrier diffusion lengths of our MOFs, superior to metal-oxo MOFs. However, unlike perovskites, the VBM in our MOFs are largely contributed by organic linkers instead of halide species, probably due to the Pb²⁺-carboxylate coordination in MOFs. Therefore, the halide species do not have an equally important role in light harvesting for our MOFs.
- The better photocatalytic performance of TMOF-10-NH₂(I) over TMOF-10-NH₂(Br) is ascribed to the enhanced carrier transport characteristics of TMOF-10-NH₂(I). The isostructural TMOFs show analogous bandgaps, but the carrier mobility and carrier diffusion lengths of TMOF-10-NH₂(I) are substantially higher than TMOF-10-NH₂(Br) (**Figure 4 and Table 1**).

3. The balance between the reduction side and oxidation side in the overall CO₂ reduction is critical. One of a key issue for the use of MOFs for CO₂ reduction is the accumulation of holes, once the electrons are consumed faster, therefore will lead to gradual self-oxidation. Although the oxidized and reduced species generated is not yet important at this initial stage, the quantitative amount will provide critical information about the charge separation and utilization. Is the consumption of electrons and holes are equivalent in terms of products? What is the cycling performance?

- We thank the reviewer for the valuable comment. First, the amounts of the oxidation products (H₂O₂) and the reduction products (CO/CH₄) are quantitatively determined

(**Figure 5a and 5e**). This indicates the consumed holes and the consumed electrons have a ratio of 0.92. The slightly lower H₂O₂ evolution rate is ascribed to the possible formation of other undetected reactive oxygen species (ROS). Second, both half photocatalytic reactions by TMOF-10-NH₂(I) are quantitatively determined for three consecutive cycles. No apparent decrease in photocatalytic activity is observed, showing the high photocatalytic stability for our MOFs (**Figure 5b and S37**). To further investigate the possible self-oxidation of photocatalysts, we have performed XPS studied for post-catalysis TMOF-10-NH₂(I) (**Figure S45**). No clear change in the binding energies of Pb, C, O, I species in TMOF-10-NH₂(I) is observed, excluding the possibility of self-oxidation of MOFs. We have added the results and discussion in the revised manuscript and ESI.

Minor suggestion:

4. *The crystal surface might also influence the catalytic performance. In the page 9, the author mentioned “the (001) facets, which is the major exposed crystal facet of TMOF-10-NH₂(I)”. What is the detailed method to determine the exposed crystal surface? How does it influence the performance? Control experiments might be necessary.*

- The exposed facets of the as-synthesized TMOF-10-NH₂(I) crystals were determined by single-crystal diffractometer and indexed by APEX-3 software. DFT calculations suggest the (001) facet of TMOF-10-NH₂ has the highest affinity towards CO₂ molecules.
- We agree with the reviewer that the exposed crystal facets of photocatalysts are critical in photocatalytic performances. However, the attempts to tune the exposed crystal facets for this class of lead halide-based MOFs were unsuccessful. If possible, we will report the influence of exposed crystal facets of TMOFs in photocatalysis in the near future.

5. *IR studies are informative to the chemical structure. It is recommended the performance of FT-IR spectra for MOFs after photocatalysis to investigate the leaching/change of organic linkers.*

- We thank the reviewer for the kind suggestion. We have performed the FT-IR spectra for TMOF-10-NH₂(I) after photocatalysis (**Figure S38**). No obvious change was observed throughout the photocatalytic process, again demonstrating no substantial change of organic linkers in our photocatalysts.

6. *For the index of PXRD pattern, hkl values are directly used without parenthesis. This is a common mistake that can even be found in some of the textbooks.*

- We thank the reviewer for the detailed comment, and **Figure 2a** has been revised accordingly.

7. *Typos in the manuscript should be fixed, eg, in the Fig. 6b, the “CO₂” was incorrectly labelled as “CO₂”.*

- We thank the reviewer for the detailed comment, and **Figure 6b** has been revised accordingly.

8. *Surface areas of these MOF should also be discussed.*

- The surface areas of TMOF-10-NH₂ are obtained by CO₂ isotherms at 273 K, using the BET theory (*J. Mater. Chem.* **2009**, 19, 2131). The surface areas have been determined to be 51.6 m²/g for TMOF-10-NH₂(I) and 53.7 m²/g for TMOF-10-NH₂(Br), respectively.

9. There are rigorous standards in the test of AQE. At least, the distance between the lamp and sample, size of the irradiated area, and temperature of the cell should be fixed before an accurate value is estimated. The AQE value reported here is not the pure MOF, but with Ru as dopant, this should be specified in the text, otherwise, it is confusing.

- We thank the reviewer for this important comment. The detailed experimental details of AQY measurement have been added into the manuscript, including the distance between the lamp and sample, the size of the irradiated area, and the temperature of the cell for AQY measurement. In this work, the lamp is fixed in 8 cm away from the catalyst membrane, and the irradiated area is measured to be 0.000625 m² (2.5 cm×2.5 cm). The temperature of the cell is fixed at 10 °C by the cooling water system during photocatalysis.
- The highest AQY value (1.36% at 400 nm) has also been specified for the TMOF-10-NH₂(I) with 1.58 wt.% Ru as cocatalyst. The main texts have been revised accordingly.

Reviewer 3

In this work, the authors report the preparation and photocatalytic application of a Metal-Organic framework comprised of Lead-Bromo/Iodo Secondary Building Units and 2-amino-teraphtalate. The authors consistently characterize the hybrid materials and thoroughly investigate their photocatalytic activity for CO₂ reduction, giving emphasis on the mechanistic aspects. This work is of great significance, showcasing how combining the chemistry of two distinct class of materials (MOFs and Organo-Lead Halide materials) can give rise to hybrid materials with enhanced properties. Such strategy can further diversify the portfolio of efficient photocatalysts and advance this field. I find this detailed research study very interesting, however there are few inconsistencies that need to be addressed. I thus recommend this manuscript for publication in Nature Communications after the following minor issues are addressed:

- We thank the reviewer for the strong endorsement.

1. *There are several spelling/grammatical mistakes in the manuscript (especially in the methods section) that need to be corrected. I would thus recommend proofreading.*

- We have carefully revised the manuscript to correct the spelling and grammatical mistakes.

2. *There are references missing at the end of the second paragraph on page 5: 'Their photocatalytic acitivity in CO₂ reduction (AQE of 1.4% at 400 nm) is the highest reported value in organolead halide hybrids when using water vapor as the sacrificial agent, and higher than many benchmark metal-oxo MOFs with the identical organic linker (e.g. UiO-66(Zr)-NH₂ and MIL-101(Fe)-NH₂) as well.'*

- We thank the reviewer for the kind comment, and have added the references accordingly (*Adv. Energy Mater.* **2022**, 12, 2004002; *Adv. Mater.* **2018**, 30, 1705512; *Adv. Sci.* **2022**, 9, 2103361).

3. *On page 4 the authors mention: 'For example, the UV-absorbing Zr₆O₄(OH)₄ clusters are the active sites for CO₂ photoreduction in the benchmark UiO-66~68 MOFs. Employing the light-absorbing linkers and/or guest molecules will introduce new charge transfer problems, such as the energetically unfavorable ligand-to-node charge transfer and noncovalent interface charge transfer, respectively.'* In the UiO-66 and Uio-68 series, the light absorption

is governed by that of the ligand, not of the SBUs. In addition, introduction of light-harvesting linkers is not problematic, but a promising strategy for enhancing the optoelectronic properties of MOF photocatalysts. I would recommend rephrasing this section, as it is very confusing.

- We thank the reviewer for this valuable suggestion, and the discussion and references have been revised as follows. “For example, the UV-absorbing $Zr_6O_4(OH)_4$ clusters are the active sites for CO_2 photoreduction in the benchmark UiO-series MOFs (*Chem Eur. J.*, **2010**, 16, 11133). The introduction of light-harvesting organic linkers and/or guest molecules are necessary to achieve visible-light-driven photocatalysis (*Chem. Soc. Rev.*, **2014**, 43, 5982; *Acc. Chem. Res.*, **2019**, 52, 356).”

4. The caption of Figure 2, should be modified, as it does not illustrate the stability, but the PXRD patterns of the as-synthesized materials.

- We thank the reviewer for the detailed comment, and the title of Figure 2 caption has been revised to “PXRD and CO_2 sorption of TMOF-10- NH_2 ”.

5. There are inconsistencies with the terminology used. More specifically, ‘RE’ should be replaced by ‘RH’ on page 2, the term ‘AQE’ should be replaced by ‘AQY’ on page 34, ‘Volts’ should be replaced by ‘eV’ on page 12, and ‘protonation’ should be replaced by ‘deprotonation’ on page 21. In addition, the authors mention on page 8 that there are ‘ $(CH_3)_2NH_2^+$ ’ guests, referring to Figure S13. Do the authors mean dimethylammonium or perhaps dimethylamine? If there are indeed cationic species, this should influence the chemical formula of the MOFs. This should be clarified and properly discussed.

- We have revised the terminologies per reviewer’s suggestions, using “RH” and “AQY” in a consistent manner.
- The typos, such as “protonation” and “volts”, have been corrected accordingly.
- Based on the X-ray crystallography studies and the vast majority of MOF synthesis using DMF, it is concluded that the framework bears an overall negative charge that are compensated by $(CH_3)_2NH_2^+$ cations residing in the porosity. This affords the chemical formula of title compounds to be $[Pb_2X]^{3+}(NH_2-bdc)_2 \cdot (CH_3)_2NH_2^+$ ($X=Br^-/I^-$). However, during the activation process in EtOH, the leaching of $(CH_3)_2NH$ is noticed, probably resulting from the proton transfer from $(CH_3)_2NH_2^+$ to amine groups (Figure S13 and S14).

6. The authors state ‘The high-energy emission is ascribed to be the charge transfer from $bdc-NH_2$ to Pb^{2+} centers, which will be discussed later.’. Please provide references of research studies in the literature or experiments/calculations supporting this argument.

- We thank the reviewer for the comment. The references have been added to this claim (*Cryst. Growth Des.* **2014**, 14, 1476; Blasse, G.; Grabmaier, B. C. Luminescent Materials; Springer Verlag: Berlin, **1994**). In addition, the DOS calculations in this work indicate that the CBM is dominated by Pb 6p orbitals and NH_2-bdc linkers contribute to VBM. Therefore, it is reasonable to attribute the high-energy emission to the charge transfer from NH_2-bdc to Pb^{2+} centers. The discussions are included on page 13.

7. The timescales for TAS and TPL are very different; ps and ns for TAS and TPL, respectively, which seems counterintuitive, as through TAS all the charge carriers can be tracked down, including the ones radiatively recombining with the holes (PL). How do the authors explain that? I would recommend a more detailed description of the insights from the TAS and TPL experiments to be added in the manuscript.

- We thank the reviewer for this important comment, and hope our additional figures, references and explanations (see below) will resolve any confusion.
- TAS equipped with different detectors are available in different time windows (*e.g.* fs-ps, ns-us), and the detection range for TAS in this study is from 0.1 to 5000 ps. Two components in the picosecond domain (τ_1 and τ_2) are noticed for TMOF-10-NH₂ (**Figure 4e**), ascribed to the the electron dynamics associated with the different electron trap states that are energetically located within the bandgap of TMOF-10-NH₂ (*Angew. Chem. Int. Ed.* **2016**, 55, 9389; *Nano Energy.* **2020**, 78, 105388). These two near-band-edge trap states accumulate the photogenerated electrons from the bottom of conduction band in a bi-exponential relaxation manner, affording two different trap depths (*J. Am. Chem. Soc.* **2015**, 137, 13440). This type of electron transfer dynamics is in the ps-ns range, which are within the scope of TAS studies.
- The lifetimes of such long-lived trap states are typically in the nanosecond domain, therefore the electron-detrapping processes are further examined by time-resolved PL spectroscopy (*Acc. Chem. Res.* **2019**, 52, 3188; *J. Am. Chem. Soc.* **2019**, 141, 13033). Both TAS and TPL are fitted using a biexponential function, showing two lifetime constants and validating our claims of the photoexcited dynamics in TMOF-10-NH₂ (**Figure S29**).
- The related figures, references and explanations have been added to the manuscript and ESI.

8. *The authors should provide the BET surface areas of the MOFs, as the high porosity of the reported materials is emphasized.*

- The surface areas of TMOF-10-NH₂ are obtained by CO₂ isotherms at 273 K, using the BET theory (*J. Mater. Chem.* **2009**, 19, 2131). The surface areas have been determined to be 51.6 m²/g for TMOF-10-NH₂(I) and 53.7 m²/g for TMOF-10-NH₂(Br), respectively.
- The low surface areas are due to the insufficient activation of MOFs, probably owing to the presence of cations in the porosity and only partial leaching of (CH₃)₂NH guests (**Figure S13 and S14**). Nevertheless, based on the CO₂ and water vapor sorption isotherms at 298 K, the gaseous CO₂ and H₂O molecules are accessible to the porosity of our lead halide-based MOFs (**Figure 2b and S16**).

REVIEWERS' COMMENTS

Reviewer #1 (Remarks to the Author):

For the revised manuscript, the authors have conducted detailed complementary and comparative experiments. All the results seem to be reasonable. I agree to be published in Nature Communication.

Reviewer #2 (Remarks to the Author):

The novelty becomes clearer after the revision and further explanation by the authors. They also made efforts to add sufficient experiments to support the finding, through the difficult time of local COVID situation. All the key concerns have been fully addressed. It is now formally recommended for publication.

Just a minor suggestion, the accuracy of BET surface area of MOFs is usually not beyond several tens of m^2/g , due to the inaccuracy in sample weighing and dead volume of the glass cell. This should be revised properly.

In the conclusion part, it is not necessary to emphasize the AQY, since Au is used as co-catalyst, and the comparison groups, lead halide perovskites, and amino functionalized metal-oxo MOFs, are quite narrow in the materials for CO_2 photoreduction. It is readily impressive to demonstrate that stable MOFs can be made from linear lead halide building blocks.

Reviewer #3 (Remarks to the Author):

The authors addressed most of my comments, however there are still several spelling mistakes that need to be corrected prior to publication (e.g., line 107 on page 6, lines 347 and 349 on page 18).